# KDM3B inhibitors disrupt the oncogenic activity of PAX3-FOXO1 in fusion-positive rhabdomyosarcoma

Yong Yean Kim [1] ✉, Berkley E. Gryder [1,2], Ranuka Sinniah[1], Megan L. Peach [3], Jack F. Shern [4], Abdalla Abdelmaksoud[5], Silvia Pomella [1,6], Girma M. Woldemichael[7,8], Benjamin Z. Stanton [1,9,10,11], David Milewski[1], Joseph J. Barchi Jr.[12], John S. Schneekloth Jr[12], Raj Chari [13], Joshua T. Kowalczyk [1], Shilpa R. Shenoy[7,8], Jason R. Evans[14], Young K. Song[1], Chaoyu Wang[1], Xinyu Wen[1], Hsien-Chao Chou [1], Vineela Gangalapudi[1], Dominic Esposito [15], Jane Jones[15], Lauren Procter[15], Maura O'Neill[16], Lisa M. Jenkins [17], Nadya I. Tarasova[18], Jun S. Wei [1], James B. McMahon[8], Barry R. O'Keefe [8,14], Robert G. Hawley [1,19] & Javed Khan [1] ✉

Fusion-positive rhabdomyosarcoma (FP-RMS) is an aggressive pediatric sarcoma driven primarily by the PAX3-FOXO1 fusion oncogene, for which therapies targeting PAX3-FOXO1 are lacking. Here, we screen 62,643 compounds using an engineered cell line that monitors PAX3-FOXO1 transcriptional activity identifying a hitherto uncharacterized compound, P3FI-63. RNA-seq, ATAC-seq, and docking analyses implicate histone lysine demethylases (KDMs) as its targets. Enzymatic assays confirm the inhibition of multiple KDMs with the highest selectivity for KDM3B. Structural similarity search of P3FI-63 identifies P3FI-90 with improved solubility and potency. Biophysical binding of P3FI-90 to KDM3B is demonstrated using NMR and SPR. P3FI-90 suppresses the growth of FP-RMS in vitro and in vivo through downregulating PAX3-FOXO1 activity, and combined knockdown of KDM3B and KDM1A phenocopies P3FI-90 effects. Thus, we report KDM inhibitors P3FI-63 and P3FI-90 with the highest specificity for KDM3B. Their potent suppression of PAX3-FOXO1 activity indicates a possible therapeutic approach for FP-RMS and other transcriptionally addicted cancers.

Rhabdomyosarcoma (RMS) is the most common soft tissue sarcoma in children, accounting for 3% of all childhood tumors[1]. There are two major subtypes, embryonal RMS and alveolar RMS. Alveolar RMS is characterized by recurrent chromosomal translocations resulting in oncogenic PAX3-FOXO1 or PAX7-FOXO1 fusion transcription factors (fusion-positive RMS, FP-RMS)[2]. Clinically, FP-RMS is associated with metastatic disease and worse overall survival[3]. Expression of PAX3-FOXO1 is crucial for the survival and proliferation of FP-RMS[4,5]. To identify drugs that inhibit the transcriptional output of PAX3-FOXO1, a

PAX3-FOXO1-selective cell-based functional reporter assay was utilized as previously described[5,6]. ChIP-seq analysis showed that a super-enhancer within the *ALK* gene locus was highly bound by PAX3-FOXO1. This *ALK* super-enhancer region was employed to express luciferase in a readout of PAX3-FOXO1-driven transcription.

Here, we use this cell-based assay to screen a 62,643-member small molecule library and identify a compound we designate P3FI-63 (PAX3-FOXO1 transcriptional Inhibitor 63) that blocks PAX3-FOXO1 downstream target gene expression. We present the structural,

biochemical, and biological characterization of P3FI-63 and a more potent analog, P3FI-90.

## Results

### Small molecule screen identifies inhibitor of PAX3-FOXO1 activity

A small molecule library of 62,643 pure compounds (NCI Molecular Targets Program) was screened at a dose of 10 μM for 24-h using a genetically engineered RH4 FP-RMS cell line expressing a PAX3-FOXO1-selective *ALK* super-enhancer-driven luciferase[6]. In parallel, we also tested the compounds on an RH4 cell line with a constitutively active CMV promoter-driven luciferase to monitor general transcriptional inhibition. We selected molecules with two characteristics: (1) Selective decrease in *ALK* super-enhancer-driven luciferase relative to CMV promoter-driven luciferase; and (2) lack of non-specific toxicity at this early timepoint as measured by 2,3-bis-(2-methoxy-4-nitro-5-sulfophenyl)-2h-tetrazolium-5-carboxanilide (XTT) cell viability assay (Fig. 1A). As described previously, we integrated the *ALK* luciferase, CMV luciferase, and XTT assays using the formula (4*ALK-Luc + 2*CMV-Luc + XTT)/7 to generate a weighted average score that resulted in the identification of 573 molecules[5]. Dose-response of the 573 molecules was obtained in a follow-up screen consisting of five 10-fold dilutions ranging from 20 μM to 2 nM. Molecules which showed proper dose-response of decreasing *ALK*-Luc with increasing dose of small molecule were chosen which further reduced the candidates to 64 compounds (Supplementary Data 1, Table 1), which were classified into 3 major groups[5]. The first group showed decreased *ALK* super-enhancer-driven luciferase concomitant with increased CMV-driven luciferase as observed with histone deacetylase inhibitors (HDACi) exemplified by N1302 (1-alaninechlamydocin)[5], designated HDACi-like. The second group showed decreased *ALK* super-enhancer-driven luciferase without a significant change in CMV-driven luciferase, which was designated *ALK*-selective. Lastly, compounds that strongly decreased both *ALK* super-enhancer and CMV-driven luciferase below 10% of control were denoted strong inhibitors (Fig. 1B). Besides N1302, there were 24 other compounds with known mechanisms of action including midostaurin (PKC-412), a previously identified inhibitor of PAX3-FOXO1, which validated our screen[7].

To investigate potential mechanisms of action of compounds without known function, we selected 4 HDAC-like, 2 *ALK*-selective, and 1 strong inhibitor with diverse molecular structures for RNA-seq analysis (Supplementary Fig. 1A; Supplementary Data 2, Tables 1–9). Gene set enrichment analysis (GSEA)[8,9] showed that P3FI-63 treatment most closely phenocopied shRNA knockdown of PAX3-FOXO1 when considering the sum of normalized enrichment scores of downregulated PAX3-FOXO1 gene sets. (Fig. 1C; Supplementary Fig. 1A; Supplementary Data 2, Tabes 1–9)[6,10]. Detailed information about the gene sets used for GSEA analyses can be found in the "Methods" section. In addition, myogenesis and apoptosis gene sets were significantly enriched, suggesting that P3FI-63 may induce differentiation and apoptosis (Fig. 1C). Consistent with this, western analysis demonstrated increased MYOG and PARP cleavage compared to DMSO control (Fig. 1D; Supplementary Fig. 1B). Direct enzymatic assay confirmed that caspase 3/7 activities were significantly increased (Supplementary Fig. 1C). Notably, P3FI-63 action did not decrease the amount of PAX3-FOXO1 protein (Fig. 1D).

The half-maximal effective concentrations (EC$_{50}$) of P3FI-63 on FP-RMS cell lines RH4, RH30, and SCMC were in the single digit μM range (Fig. 1E). Next, we tested other pediatric solid cancer lines to determine broader applicability of P3FI-63. EC$_{50}$ values for fusion negative RMS cell lines (RD: 7.16 μM and CTR: 2.92 μM), Ewing's sarcoma (TC-32: 1.7 μM and A673: 3.7 μM) and osteosarcoma (OSA: 7.5 μM and HU09: 3.1 μM) as well as primary human fibroblast (7250: 6.62 μM) were also in the single digit μM range (Supplementary Fig. 1D). We also tested

P3FI-63 on the NCI-60 cancer cell lines[11], which showed that P3FI-63 was mostly cytostatic in adult cancer lines (Supplementary Fig. 2). Given that P3FI-63 downregulated the target genes of PAX3-FOXO1 in FP-RMS and had broad activity in other pediatric solid tumor cell lines, we set out to determine its molecular target(s).

### P3FI-63 targets histone lysine demethylases

First, we examined the GSEA results focusing on enriched gene sets following treatment with P3FI-63[8,12]. Remarkably, among the most highly upregulated gene sets by P3FI-63 were genes associated with JIB-04 and GSK-J4 treatment (Fig. 1F)[13,14]. JIB-04 and GSK-J4 are inhibitors of the Jumonji domain (JmjC) family of histone lysine demethylases (KDMs) that are involved in epigenetic control of transcription. This suggested that P3FI-63 may act by inhibiting KDMs. In support of this notion, submission of P3FI-63 to the Similarity Ensemble Approach (SEA) Search Server (https://sea.bkslab.org/) to generate predictions for potential targets revealed that P3FI-63 could be a ligand for multiple KDMs (Supplementary Data 1, table 2)[15,16]. Next, we performed ATAC-seq after treatment with P3FI-63 vs DMSO to assess new open chromatin regions using GSEA. We found that there was an enrichment for a gene set induced by the KDM1A inhibitor SP2509 (FDR = 0.097; Supplementary Fig. 3A; Supplementary Data 1, table 3)[17]. Therefore, we hypothesized P3FI-63's mechanism of action to be the inhibition of KDMs. Although KDMs were most consistently predicted as P3FI-63 targets by RNA-seq, GSEA, and ATAC-seq, other epigenetic modifiers were suggested including HDACs and PRMT5. Hence, we further assessed the direct inhibition of KDMs by P3FI-63 as well as HDACs and PRMT5 using cell-free in vitro enzyme inhibition assays. Confirming our hypothesis, P3FI-63 inhibited multiple KDMs including KDM3B, KDM4B, KDM5A, and KDM6B, with highest potency against KDM3B at a half-maximal inhibitory concentration (IC$_{50}$) of 7 μM (Fig. 1G). In contrast, P3FI-63 had no inhibitory activity against HDAC1, HDAC2, HDAC3, or PRMT5. Next, we performed RNA-seq analysis after treatment with known KDM inhibitors, JIB-04, GSK-J4, and GSK690 to determine if KDM inhibition can result in significant downregulation of PAX3-FOXO1 target genes. GSEA analysis showed that the multi-KDM inhibitors JIB-04 and GSK-J4 downregulated a subset of the compiled PAX3-FOXO1 signatures (Supplementary Fig. 3B; Supplementary Data 1, Tables 4–6). By comparison, GSK690, a KDM1A inhibitor, downregulated only one PAX3-FOXO1-associated gene set. To validate P3FI-63's activity on KDMs, we performed western analysis of P3FI-63-treated FP-RMS cell lines for changes in methylation status of histone 3 lysines: H3K9 (demethylated by KDM3B, KDM4B, KDM1A), H3K4 (KDM1A, KDM5A) and H3K27 (KDM6B). After 24-h treatment of RH4 and RH30 cells with P3FI-63, we found significant increases in the methylation of all three lysine substrates compared to DMSO, confirming that multiple histone demethylases are targeted by P3FI-63 in FP-RMS cells (Fig. 1H; Supplementary Fig. 3C). Given the semi-selectivity of P3FI-63 towards KDM3B, we next investigated KDM3B expression levels in RMS cell lines using the Broad Institute's Dependency Map (DepMap, https://depmap.org/portal/) portal. We observed that KDM3B is significantly expressed in RMS cell lines, having the second highest levels of KDM3B expression overall compared to other cancer lines, indicating that it is a potential target for therapy (Supplementary Fig. 3D).

### Identification of chemical analog of P3FI-63 with improved properties

P3FI-63 had low solubility in water with a partition coefficient logP of 2.36 making preclinical in vivo testing a challenge. Hence, we conducted a molecular similarity search with P3FI-63 in the Enamine catalog and identified 26 commercially available compounds of similar structure (Supplementary Data 1, Table 7). Performing the *ALK* super-enhancer-driven luciferase screen with the 26 compounds identified one new compound designated as P3FI-90 (Compound 26)

(Fig. 2A, Supplementary Fig. 4), which showed significant inhibition of PAX3-FOXO1 driven luciferase (Supplementary Data 1, Table 7). P3FI-90 had increased water solubility (logP of 0.82) and showed low to sub-micromolar EC$_{50}$ activity in FP-RMS (Fig. 2B). P3FI-90 was also tested on the NCI-60 panel of adult cancer cell lines which again

showed mostly cytostatic effects except for renal cancer cell lines where cell killing was observed at the highest dose tested (Supplementary Fig. 5).

Similar to P3FI-63, cell-free in vitro enzyme inhibition assays confirmed that P3FI-90 had highest inhibitory selectivity towards

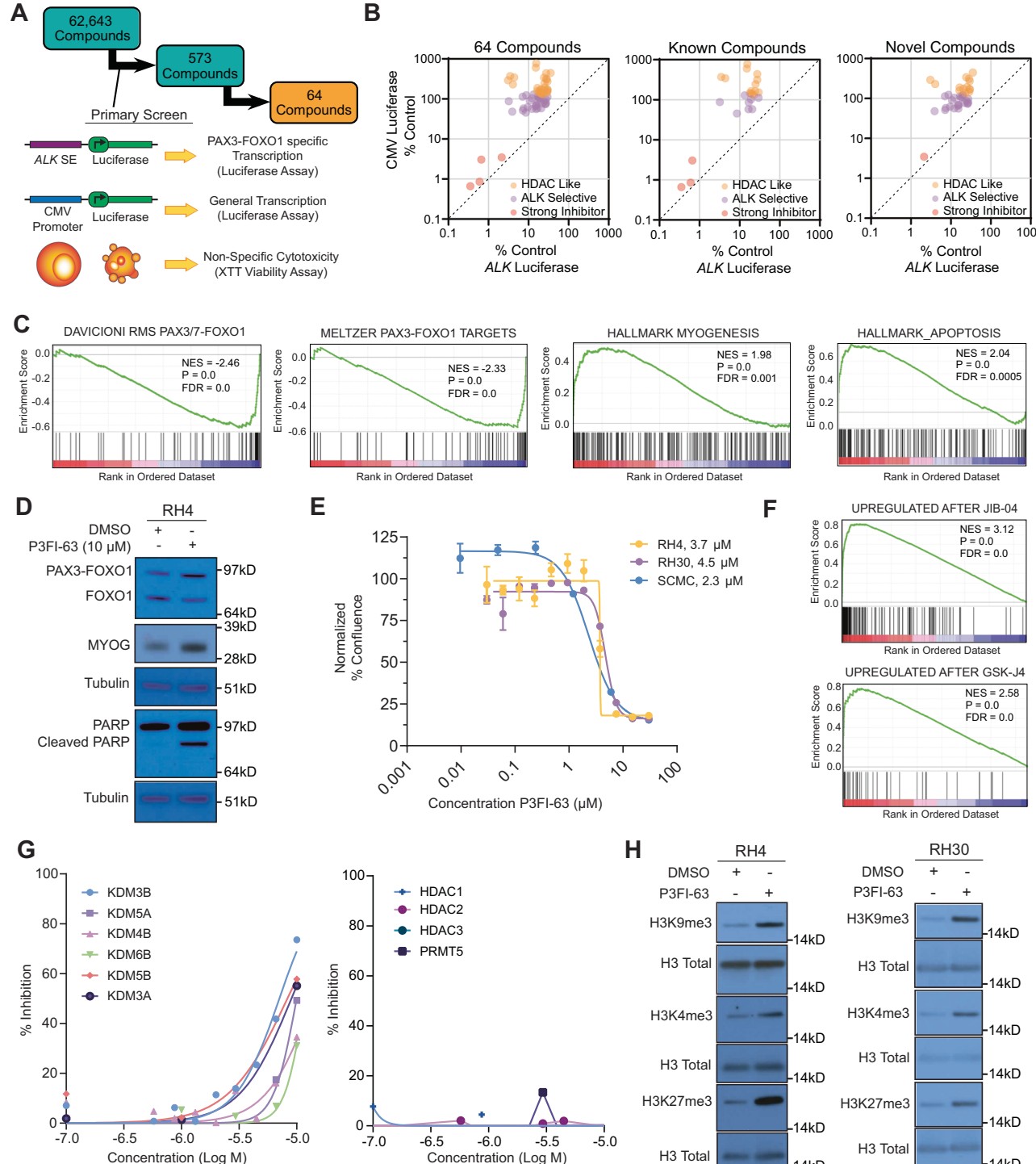

**Fig. 1 | Identification and characterization of P3FI-63. A** Small molecule screen strategy to identify compounds that selectively inhibit PAX3-FOXO1-driven transcription **B** Top 64 compounds and their inhibitory characteristics of *ALK* superenhancer-driven luciferase vs. CMV-driven luciferase (20 μM). **C** RNA-seq GSEA analysis of RH4 cells treated with compound P3FI-63. Normalized Enrichment Score (NES), False Discovery Rate (FDR). **D** Western validation of RNA-seq showing upregulation of MYOG and PARP cleavage (*n* = 1). **E** EC$_{50}$ calculations of P3FI-63 on

PAX3-FOXO1 positive RMS cell lines RH4, RH30, and SCMC. Data presented as mean values ± SEM. *n* = 3 biological replicates, error bar = Standard Error. **F** RNA-seq GSEA analysis of P3FI-63-treated RH4 cells reveal KDMs as possible targets. **G** Direct enzymatic inhibition assays of P3FI-63 against KDMs, HDACs, and PRMT5. Single experiment. **H** Western validation of KDM inhibition by P3FI-63 showing increases in methylation of H3K9me3, H3K4me3, and H3K27me3 (*n* = 1). Source data are provided as a Source Data file.

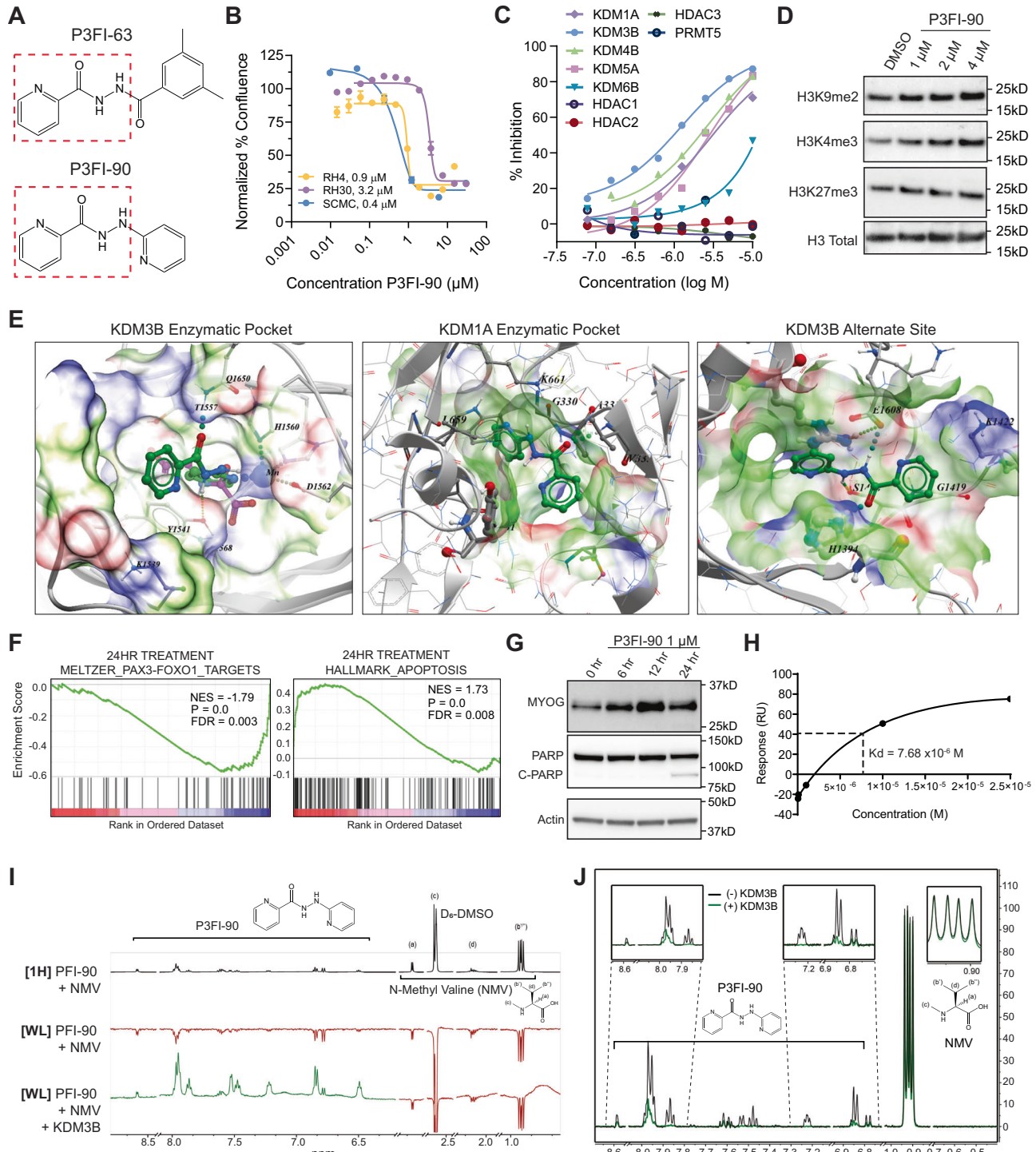

**Fig. 2 | Characterization of P3FI-90. A** P3FI-90 structure compared to P3FI-63. **B** EC$_{50}$ calculations for P3FI-90 on PAX3-FOXO1 positive RMS cell lines RH4, RH30, and SCMC. Data presented as mean values ± SEM. $n = 3$ biological replicates, error bar = Standard Error. **C** Direct enzymatic inhibition assays of P3FI-90 against KDMs, HDACs, and PRMT5 at 8 concentrations. Single Experiment. **D** Western validation of KDM inhibition showing increases in H3K9 and H3K4 methylation. One representative experiment from $n = 2$ is shown. **E** Left panel: Binding site of P3FI-90 (green) in KDM3B (PDB: 4c8d) overlaps with the binding site of the cofactor, 2-oxoglutarate (magenta). Middle panel: P3FI-90 docks well into the active site of KDM1A/LSD1 (PBD: 2Z3Y) forming hydrogen bonds with the main chain NH-groups of A331 and V-333. Pyridine moieties fit into hydrophobic pockets formed by the site chains of V-333, T335, A806, Tyr 761, L659, and Trp 751. Right panel: P3FI-90 alternative binding site on KDM3B that overlaps with N-phenyl-N′-pyridin3-yl-urea on KDM3B structure PDB:5RAZ and that is structurally similar to P3FI-90. The

pocket is on the opposite side of the active site. In the alternative binding site, P3FI-90 forms hydrogen bonds with Ser1418, His 1394, and Glu1608. **F** RNA-seq GSEA analysis of RH4 cells treated with P3FI-90 showing downregulation of PAX3-FOXO1 target gene set, and upregulation of Apoptosis gene set. **G** Western validation of RNA-seq showing upregulation of MYOG and PARP cleavage. One representative experiment from $n = 2$ is shown. **H** Surface plasmon resonance show binding of P3FI-90 to KDM3B. KD of $7.68 \times 10^{-6}$ M, R$_{max}$ of 130.5 RU, and Chi$^2$ of 4.78. **I** Ligand-observed nuclear magnetic resonance (NMR) technique WaterLOGSY (WL) with positive peaks showing P3FI-90 binding to KDM3B when KDM3B is present. N-Methyl-L-Valine (NMV) as non-binding control. **J** Ligand-observed NMR technique Carr-Purcell Meiboom-Gill (CPMG) showing attenuation in P3FI-90 peaks in the presence of KDM3B indicative of binding. Source data are provided as a Source Data file.

KDM3B (Fig. 2C). Given that KDM1A was identified as a possible target of P3FI-63 by ATAC-seq and RNA-seq, KDM1A was also included in the enzyme inhibition assay. Interestingly, P3FI-90 inhibited KDM1A although it is not a member of the JmjC family of KDMs and uses flavin adenine dinucleotide (FAD) as a cofactor instead of α-ketoglutarate. Western analysis showed an increase of H3K4me3 and H3K9me2 but only mild increase of H3K27me3 at doses up to 4 μM (Fig. 2D; Supplementary Fig. 6A). This was consistent with P3FI-90 having higher inhibitory potency for KDM3B, KDM4B, KDM5A, and KDM1A which target H3K9 (KDM3B, KDM4B, KDM1A) and H3K4 (KDM1A, KDM5A) than KDM6B (H3K27). Next, we performed in silico docking experiments to determine if P3FI-90 possessed suitable chemical characteristics to bind to KDMs. Docking of P3FI-90 in KDM3B structures from the Research Collaboratory for Structural Bioinformatics Protein Data Bank (PDB):4C8D, 5R7X, 5RAA, 5RAM, 5RAN, 5RAO, 5RAZ and 6RBJ produced essentially identical compound poses in the active site where it interacted with Mn2+ through a pyridinyl moiety, similar to the ligands that were present in 5RAM, 5RAN, 5RAO and 6RBJ (Fig. 2E). It also formed hydrogen bonds with the side chains of Gln1650, Tyr1541 and Asp1562. Docking analysis suggested that P3FI-90 interacts with metal ions in the active sites of KDM4B, KDM5A, KDM5B and KDM6A, but not KDM6B (Supplementary Fig. 7). Docking analysis based on Radial Convolutional Neural Net (RTCNN) score, which integrates factors determining binding energy, showed a favorable negative score of −25.4 for KDM3B (PBD: 4C8D) (Supplementary Data 1, Table 8). Only KDM6B (PBD: 5FP3) had a lower score of −29.6 but did not show interaction with the metal ion indicating that the docking is likely unfavorable. This was in line with the enzyme inhibition assay in which P3FI-90 showed less inhibition of KDM6B compared to KDM3B (Fig. 2C). Also, hydrogen bond interactions were most extensive for KDM3B and less extensive in other KDMs which correlates with higher affinity towards KDM3B. Next, given the unexpected finding that P3FI-90 also inhibited KDM1A, we performed docking analysis on KDM1A which showed a favorable RTCNN score of −23.7 with the active site of KDM1A in agreement with the enzyme inhibition assay (Fig. 2E; Supplementary Data 1, Table 8). As expected, the mode of interaction was different from that of other KDMs given that KDM1A is not a member of the JmjC family of demethylases. Finally, we were interested to see if P3FI-90 could bind to KDM3B outside the enzymatic pocket. We found that P3FI-90 has an alternative binding site that overlaps with the binding pocket occupied in the KDM3B structure PDB:5RAZ by N-phenyl-N′-pyridin3-yl-urea that is structurally similar to P3FI-90 and may potentially impact enzymatic activity allosterically.

Next, we performed RNA-seq analysis 24 h after P3FI-90 and P3FI-63 treatment as in the drug screen. GSEA showed significant enrichment of multiple apoptosis gene sets along with downregulation of PAX3-FOXO1 targets (Fig. 2F; Supplementary Data 3, Tables 1–2). Interestingly, the myogenesis gene set that was upregulated by P3FI-63 at 6-h treatment (Fig. 1C) was absent at 24 h indicating that induction of myogenic differentiation by P3FI-63 and P3FI-90 may be an early event. To better understand the time course of myogenesis, we performed Western analysis of MYOG, a master regulator of muscle differentiation, which showed that at 6 and 12 h there was an increase followed by a sharp decrease at 24 h consistent with the GSEA analysis (Fig. 2G; Supplementary Fig. 6B). Additionally, there was initiation of PARP cleavage at 24 h, indicating apoptosis, consistent with the GSEA prediction. We also quantitated caspase 3/7 activity for apoptosis and found robust increase in caspase activity after P3FI-90 treatment (Supplementary Fig. 6C). Lastly, we examined the transcript levels of the KDMs after treatment of P3FI-63 and P3FI-90 and found no significant changes compared to DMSO treatment, indicating that the biological effect is due to enzymatic inhibition and not due to downregulation of KDM transcription (Supplementary Fig. 6D).

## Biophysical interaction between P3FI-90 and KDM3B

To establish that P3FI-90 directly binds to KDM3B, we used Surface Plasmon Resonance (SPR)[18] to determine binding of P3FI-90 to synthesized full-length KDM3B which showed robust enzymatic activity in demethylating H3K9me2 (Supplementary Fig. 8A). SPR demonstrated binding of P3FI-90 to KDM3B with a $K_d$ of $7.68 \times 10^{-6}$ M (Fig. 2H). We also performed two bio-orthogonal ligand-binding detection experiments using nuclear magnetic resonance (NMR) techniques. For the NMR experiments, we employed the truncated KDM3B protein previously used to determine the crystal structure of the KDM3B enzymatic pocket[19]. We initially obtained standard proton ($^1$H), carbon ($^{13}$C), and HSQC NMR spectra of samples containing P3FI-90 and 2-OG alone and together in buffer, as well as in the presence or absence of the KDM3B (Supplementary Fig. 8B). These data provided unambiguous peak assignments for P3FI-90. The assignments served as a reference used in the ligand-observed NMR experiments. We first subjected these samples to Water Ligand-Observed via Gradient Spectroscopy (WaterLOGSY) $^1$H NMR[20]. We observed positive proton peaks for P3FI-90 only in the presence of KDM3B indicating protein binding (Fig. 2I). When KDM3B was absent, P3FI-90 had negative peaks indicating lack of binding as expected. Our non-binding negative control N-Methyl-L-Valine (NMV) showed negative peaks both with and without KDM3B.

Next, we subjected the samples to Carr-Purcell-Meiboom-Gill Relaxation-Editing $^1$H NMR (CPMG), an alternative ligand-observed NMR method, to confirm binding[21]. CPMG analysis of P3FI-90 showed binding of P3FI-90 to KDM3B indicated by the attenuation of the P3FI-90 peaks when compared to P3FI-90 without KDM3B (Fig. 2J). Again, our non-binding negative control molecule NMV showed no binding by CPMG as indicated by lack of attenuation of NMV peaks in the presence of KDM3B. Interestingly, nearly all $^1$H signals of P3FI-90 showed attenuation to ~75%, indicating that the vast majority of the P3FI-90 structure binds to the catalytic domain of KDM3B protein in solution. Hence, we confirmed that P3FI-90 binds to KDM3B in solution by three orthogonal methods.

## KDM3B is essential for survival in FP-RMS

Given that P3FI-63 and P3FI-90 target KDM3B, we investigated whether KDM3B was essential for survival of FP-RMS. We developed a tiled CRISPR library with sgRNA designed across all *KDM3B* exons. As controls, we used sgRNA for 8 essential genes including *PAX3-FOXO1*, *MYOD*, and 20 non-targeting sgRNA from the Brunello CRISPR library[22]. We found that 7.1% of sgRNAs which target *KDM3B* were depleted by >4-fold in the RH4 FP-RMS cell line with hot spots in the enzymatic domain of *KDM3B*, indicating that it is in fact essential for survival of FP-RMS (Fig. 3A). It is noteworthy that *KDM3B* has not been recognized as a dependent gene in FP-RMS (e.g., in DepMap), most likely due to a limited selection of sgRNAs against this gene.

## KDM knockdown mirrors the P3FI-90-induced transcriptome

To demonstrate that P3FI-90 downregulation of PAX3-FOXO1 target genes is through the inhibition of KDMs, we used the genetic knockdown strategy via CRISPRi. We targeted the top four KDMs inhibited by P3FI-90, *KDM3B*, *KDM1A*, *KDM4B*, and *KDM5A*. We achieved effective knockdown of KDMs measured at the transcript level and at the protein level (Fig. 3B, C; Supplementary Fig. 9A–C). RNA-seq analysis followed by GSEA showed that *KDM3B* knockdown significantly downregulated 9 of 12 PAX3-FOXO1 target gene sets that were downregulated by P3FI-90 compared to 6 of 12 for *KDM1A* knockdown and 5 of 12 for *KDM5A* (Fig. 3D; Supplementary Data 3, Tables 3–10). Comparison of KDM knockdown to *PAX3-FOXO1* knockdown using shRNA showed that *KDM3B* knockdown decreased PAX3-FOXO1 gene sets most robustly (Supplementary Fig. 9D). However, we did not see enrichment of the upregulated gene sets observed following P3FI-90 treatment, in particular, those involved in apoptosis. Given that

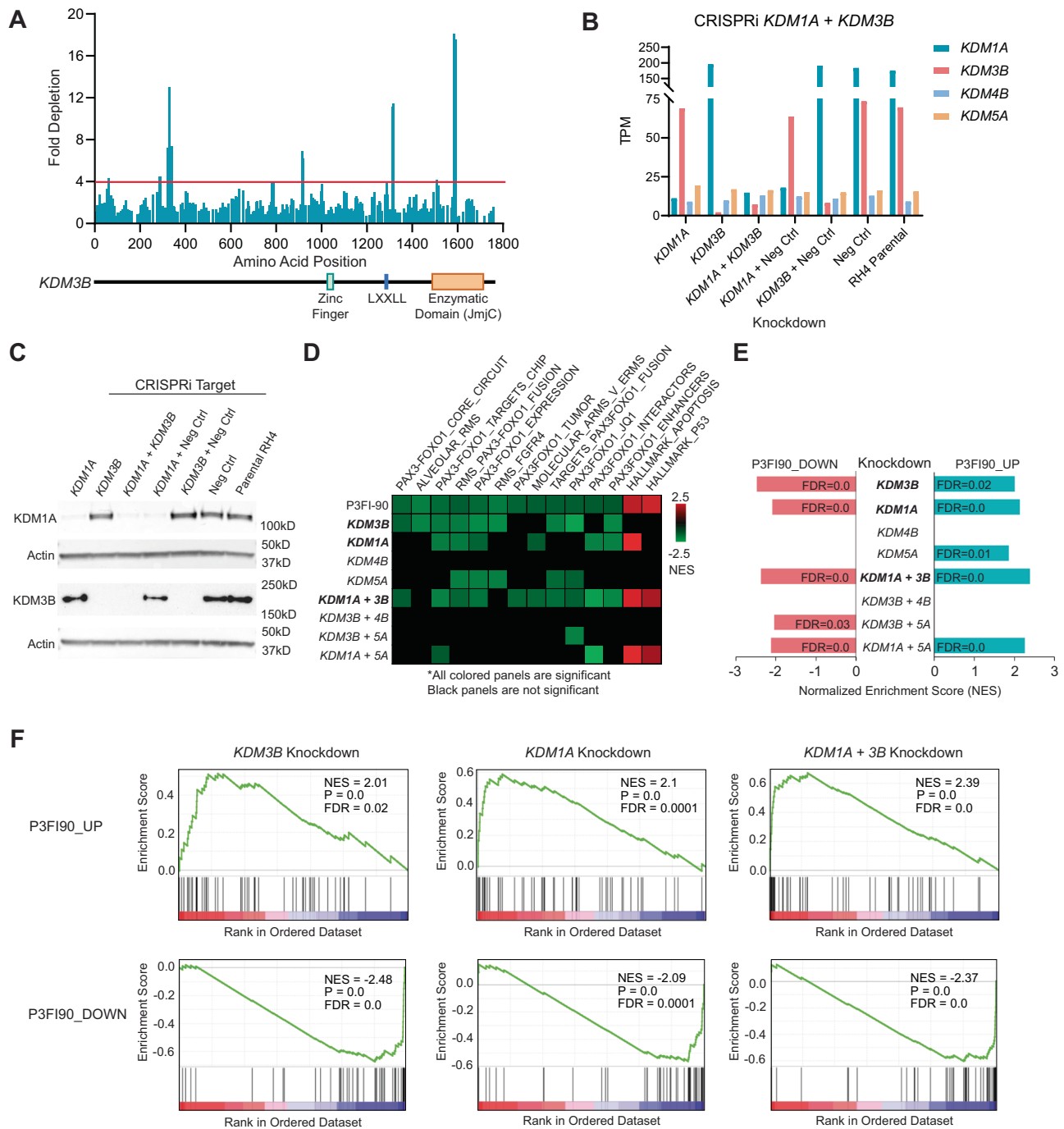

**Fig. 3 | CRISPR-based characterization of KDMs. A** CRISPR knockout with sgRNAs tiled across the entire *KDM3B* coding region. Analysis of fold depletion of sgRNAs as walking average of 4 sgRNAs. **B** RNA-seq TPM of KDMs after CRISPRi knockdown of KDMs. **C** Western validation of CRISPRi knockdown of KDMs. One representative experiment from *n* = 2 is shown. **D** GSEA analysis of RNA-seq data by P3FI-90 and KDM knockdowns. Refer to Supplementary Data 3, Tables 3–10 for Normalized Enrichment Scores (NES), *p*-values, and False Discovery Rates (FDR). **E** GSEA using P3FI90_UP and P3FI90_DOWN gene sets created using top 50 differentially expressed transcripts and **F** GSEA plots (Gene List on Supplementary Data 7, Table 3). Source data are provided as a Source Data file.

inhibition of KDM3B leads to increases in the repressive mark H3K9me2, lack of upregulated gene sets was in line with KDM3B function. Since H3K4 methylation, a marker of transcriptional activation, was also increased we investigated upregulated GSEA gene sets in *KDM1A* and *KDM5A* knockdowns. Interestingly, we found enrichment for apoptosis following *KDM1A* knockdown but not in *KDM5A* knockdown. *KDM1A* knockdown also downregulated PAX3-FOXO1 gene sets although not as robustly as *KDM3B* knockdown. The combined knockdown of *KDM1A* and *KDM3B* showed enrichment for apoptosis

gene sets together with downregulation of PAX3-FOXO1 targets, most closely recapitulating the P3FI-90-induced transcriptome in RH4 cells (Fig. 3D). Next, to assess the P3FI-90 regulated transcripts at the gene level, we took the top 50 most differentially upregulated and downregulated genes after P3FI-90 treatment vs DMSO and created new gene sets P3FI90_UP and P3FI90_DOWN respectively (Supplementary Data 7, Table 3). We found that enrichment for P3FI90_DOWN was most negative after *KDM3B* knockdown while P3FI90_UP was most positive after *KDM1A* knockdown (Fig. 3E, F). Moreover, P3FI90_UP and

P3FI90_DOWN were the most significantly enriched after combined *KDM1A* plus *KDM3B* knockdown, indicating that P3FI-90 is best genocopied by the combination of KDM3B and KDM1A inhibition.

## scRNA-seq shows P3FI-90-induced decrease in PAX3-FOXO1 signature, myoblast signature, and G2/M population

In order to study the effects of P3FI-90 at the single-cell level, we performed scRNA-seq analysis after treating cells with P3FI-90 for 16 h, a time point before apoptosis. Integrating the P3FI-90 and DMSO samples together, Cluster 2 showed the largest fold difference in cell number between the P3FI-90 and DMSO groups with 82.0% of the cluster comprised of DMSO-treated cells (Fig. 4A, B). Cluster 2 was also the largest cluster of DMSO-treated cells overall and hence represented a major population of RH4 cells during exponential growth. Clusters 3 and 8 showed the greatest fold change following P3FI-90 treatment compared to DMSO. To characterize the clusters, we examined myogenic differentiation using the top 50 differentially expressed genes from a previously published scRNA-seq profile (Supplementary Data 4, Tables 1–2)[23]. RMS cells are actively dividing in culture, resembling highly proliferative myoblast cells. The myoblast signature for Cluster 2 was higher than Clusters 1, 3, 6, and 8 (Fig. 4C; Supplementary Data 4, Table 3) whereas the less proliferative myocyte signature was higher in Cluster 6 compared to the rest of the clusters. Differential gene expression analysis showed Cluster 6 with high expression of skeletal muscle genes including multiple myosin light chain and heavy chain genes (Supplementary Data 4, Tables 4–11). Clusters 3 and 8 which increased the most after P3FI-90 treatment had reduced expression of myoblast genes indicating that P3FI-90 treatment led to an increase in the number of cells with the lowest myoblast signature. When we examined the PAX3-FOXO1 signature[24], we found that Clusters 3 and 8 also had lower expression of PAX3-FOXO1 downstream targets. Heatmap hierarchical clustering using leading edge genes from GSEA analysis of myoblast signatures and PAX3-FOXO1 signature both showed that Clusters 3 and 8 separated from the rest of the Clusters (Supplementary Fig. 10A). These findings demonstrated a strong correlation between PAX3-FOXO1 expression and maintenance of the myoblast signature. We next performed cell cycle analysis of the scRNA-seq clusters (Fig. 4D, E). Overall, there was an increase of cells in G1 and S while G2M was decreased after P3FI-90 treatment, indicating potential G2M arrest. When examining which clusters were associated with the decrease in G2M, we saw a decrease in Cluster 2 after P3FI-90 treatment (Fig. 4E). To better understand the cell cycle changes, we performed flow cytometry analysis at 24 h after treatment with P3FI-90 vs DMSO in the SCMC FP-RMS cell line. We found profound accumulation of S phase cells while G2 and M were decreased, indicating S phase blockade (Fig. 4F). These findings indicated that P3FI-90 treatment resulted in decreased expression of PAX3-FOXO1 downstream targets concomitant with decreased myoblast signature and cell division.

## P3FI-90 increases H3K4 and H3K9 methylation

To determine how inhibition of KDM3B results in abrogation of PAX3-FOXO1 action, we performed ChIP-seq analysis using antibodies against PAX3-FOXO1, H3K4me3, H3K9me2, H3K27me3, and H3K27ac. First, we performed ChromHMM analysis using our previously published ChIP-seq data to assign the chromatin into 16 states[6,24]. We observed that H3K9me2 was increased throughout the entire genome after treatment with P3FI-90 compared to DMSO, with the most pronounced increase at enhancer regions (Fig. 5A). ChromHMM also showed that H3K4me3 was increased mostly in the promoter region which is consistent with its function. Next, we examined transcription start sites (TSS) and PAX3- FOXO1 binding sites. We found that there was an increase in histone methylation at H3K9me2 both at the TSS and PAX3-FOXO1 binding sites in P3FI-90 treated cells compared to DMSO (Fig. 5B). This was consistent with P3FI-90 having the highest

inhibition of KDM3B. There was also an increase in H3K4me3 around TSS following treatment with P3FI-90, consistent with P3FI-90 inhibiting KDM1A and KDM5A which demethylate H3K4. We saw no changes in the methylation status of H3K27. These changes corresponded with P3FI-90 selectivity for each of the respective histone marks.

Since PAX3-FOXO1 drives core regulatory transcription at super-enhancer sites marked with H3K27ac, we examined if there were any changes to H3K27ac around the PAX3-FOXO1 sites. There was no change in H3K27ac at TSS or PAX3-FOXO1 sites with treatment of P3FI-90 compared to DMSO (Fig. 5B). Super-enhancer ROSE analysis also showed no changes in super-enhancers and no changes at all enhancer sites when RH4 cells were treated with P3FI-90 compared to DMSO (Supplementary Fig. 11A). P3FI-90 also did not change the binding of PAX3-FOXO1 as assessed by ChromHMM (Fig. 5A) and genome-wide profile (Fig. 5B) despite increases in H3K9me2, consistent with a previous report of PAX3-FOXO1 having the ability to bind heterochromatin as a pioneering factor[25]. We also examined the protein level of PAX3-FOXO1 by Western since PAX3-FOXO1 binds super-enhancers and found that P3FI-90 did not directly decrease PAX3-FOXO1 protein levels (Supplementary Fig. 11B). These results are similar to the Western blot data showing little to no change in PAX3-FOXO1 after P3FI-63 treatment (Fig. 1D) and consistent with the P3FI-90 ChIP-seq data (Fig. 5B; Supplementary Fig. 11C).

Given that H3K9me2 was increased most significantly at enhancer regions, we wanted to know the relative distribution of H3K9me2 at PAX3-FOXO1 binding sites compared to other transcription factors. We performed H3K9me2 ChIP-seq in triplicates after P3FI-90 vs DMSO treatment and used differential peak analysis which identified 621 upregulated peaks and 0 downregulated peaks. Nearest gene analysis found 245 unique genes (Supplementary Data 5, Table 1). We then performed ChIP Enrichment Analysis (ChEA) via a web-based analysis tool Enricher (https://maayanlab.cloud/Enrichr/) which compares gene lists from previously published ChIP-seq of transcription factors (Fig. 5C; Supplementary Data 5, Tables 1–2)[26]. PAX3-FOXO1 was the top-ranked transcription factor that overlapped with H3K9me2 (Supplementary Data 5, Table 2). Next, we performed H3K4me3 ChIP-seq in triplicates and differential peak analysis after treatment with P3FI-90 vs DMSO. There were 2399 upregulated peaks and 0 downregulated peaks. Nearest gene analysis identified 1812 unique genes of which we took the top 20% of the most differentially upregulated genes and performed gene ontology analysis using the web-based tool (https://geneontology.org/)[27]. Gene ontology analysis showed that anoikis, a form of apoptosis, was the top biological process with 22.8 fold enrichment and that many other apoptosis processes were also significantly enriched (Fig. 5D; Supplementary Data 5, Tables 3–4). Therefore, these ChIP results validated the bulk RNA-seq data; notably, that P3FI-90 suppresses PAX3-FOXO1 function concomitant with induction of apoptosis.

We also assessed if P3FI-90 would affect binding of KDM3B to target chromatin. ChIP-seq analysis of KDM3B showed that there was minimal change to KDM3B binding to TSS or PAX3-FOXO1 sites when treated with PFI-90 vs DMSO (Supplementary Fig. 11D). This indicated that P3FI-90 does not change KDM3B binding but inhibits its enzymatic function. Next, given that H3K9me2 is a heterochromatin mark, we looked to see if another heterochromatin mark HP1 was increased after P3FI-90 treatment. ChIP-seq of HP1 showed that although there was no change in HP1 at TSS, there was an increase at PAX3-FOXO1 sites after treatment with P3FI-90 vs DMSO (Supplementary Fig. 11E). This indicates that P3FI-90 treatment increased the heterochromatin character of PAX3-FOXO1 sites. This finding is consistent with H3K9me2 being a heterochromatin mark, and a correlation analysis between H3K9me2, HP1, and KDM3B showed that indeed they were colocalized (Supplementary Fig. 11F).

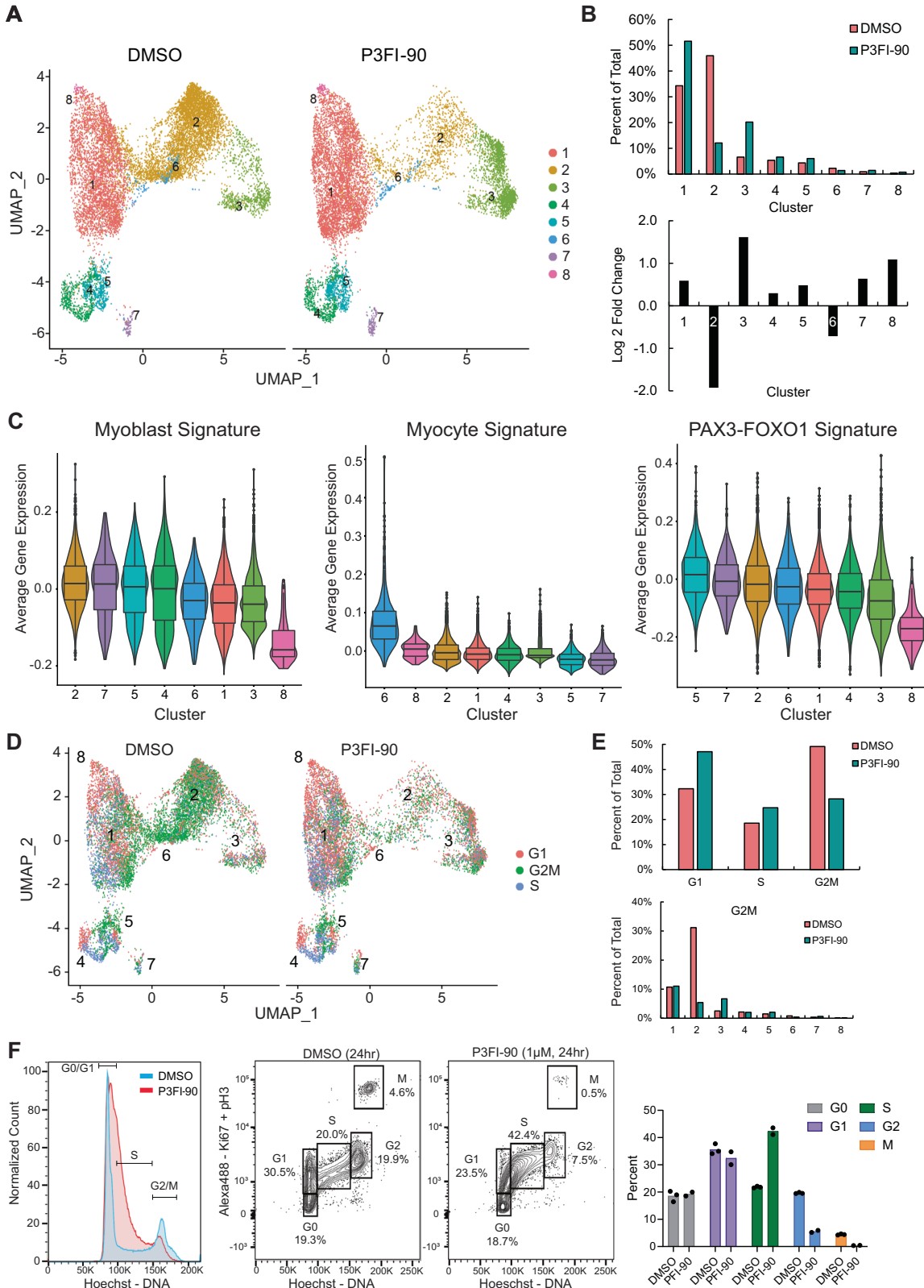

**Fig. 4 | scRNA-seq analysis of P3FI-90. A** UMAP of clusters after treatment with DMSO vs P3FI-90 for 16 h. **B** Cells in clusters as percent of total and as Log 2-fold change for P3FI-90 compared to DMSO. **C** Clusters ordered by average expression of Myoblast genes, Myocyte genes, and PAX3-FOXO1 target genes. *n* = 1 independent experiment; box plots of median and quartiles, whiskers showing

1.5 × interquartile ranges. Statics from GSEA analysis and adjusted for multiple testing by GSEA. **D** UMAP of clusters by cell cycle. **E** percentage quantitation of cell cycle overall, and G2M across clusters. **F** Flow cytometry of cell cycle in SCMC cell line after 24 h treatment with P3FI-90 at 1 μM. DMSO *n* = 3 biological replicates, P3FI-90 *n* = 2 biological replicates. Source data are provided as a Source Data file.

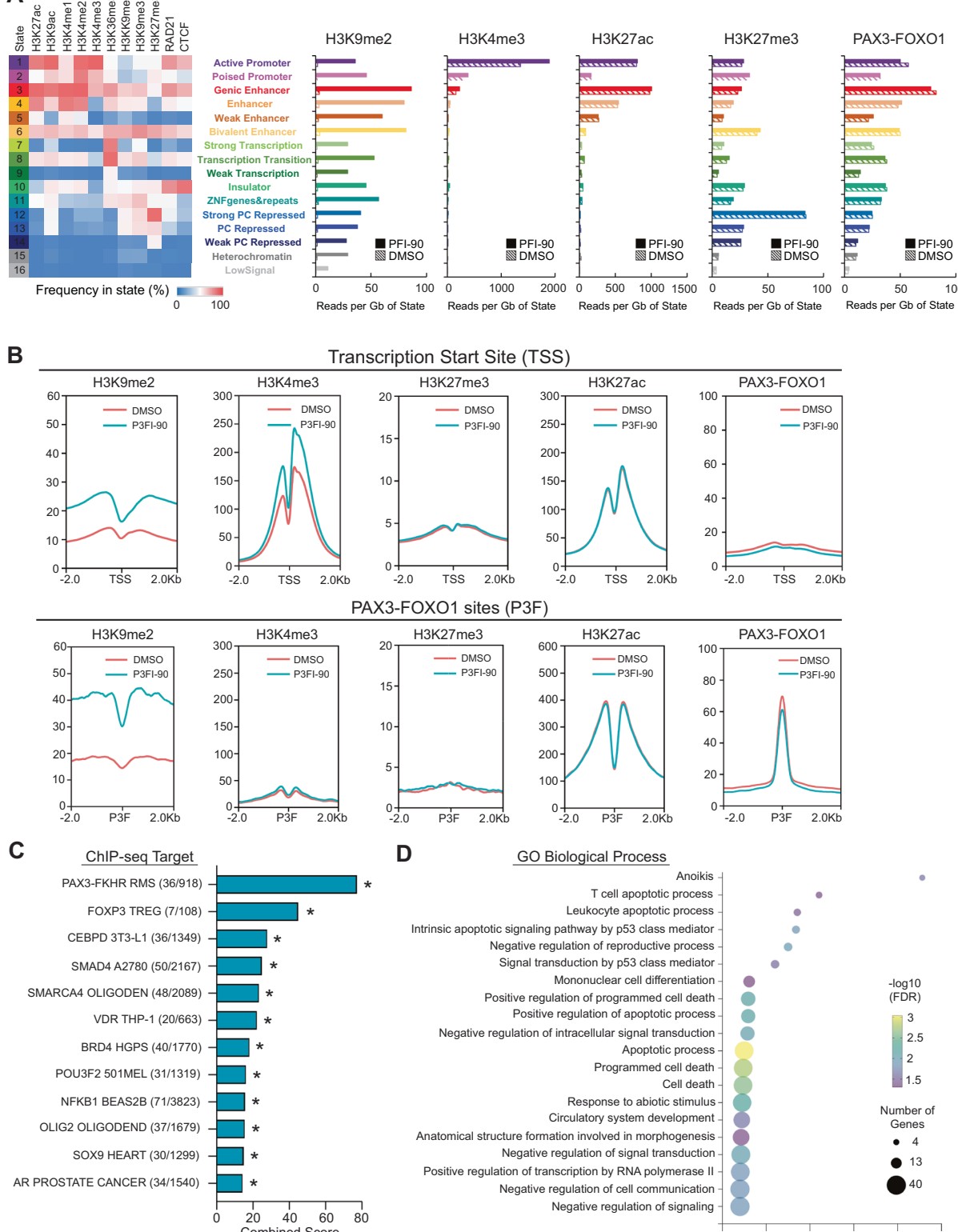

**Fig. 5 | Histone methylation profile of P3FI-90. A** ChromHMM analysis of read counts at chromatin states for H3K4me3, H3K9me2, H3K27me3, H3K27ac, and PAX3-FOXO1. $n = 1$. **B** ChIP-seq of H3K4me3, H3K9me2, H3K27me3, H3K27ac, and PAX3-FOXO1 genome-wide analysis at transcription start sites (TSS) and PAX3-FOXO1 (P3F) binding sites. Showing one representative profile from $n = 3$ except H3K27me3 which is $n = 1$. **C** ChIP enrichment analysis (ChEA) of H3K9me2

differential peaks in RH4 cell line treated for 24 h with P3FI-90 (1 μM) vs DMSO, $n = 3$. *$p$ = values can be found in supplementary Data 5, Table 2. ChEA analysis uses Fisher's exact test. **D** Gene ontology analysis of H3K4me3 differential peaks (top 20%) 24 h after treatment with P3FI-90 (1 μM) vs DMSO, $n = 3$. Gene ontology uses Fisher's exact test. Source data are provided as a Source Data file.

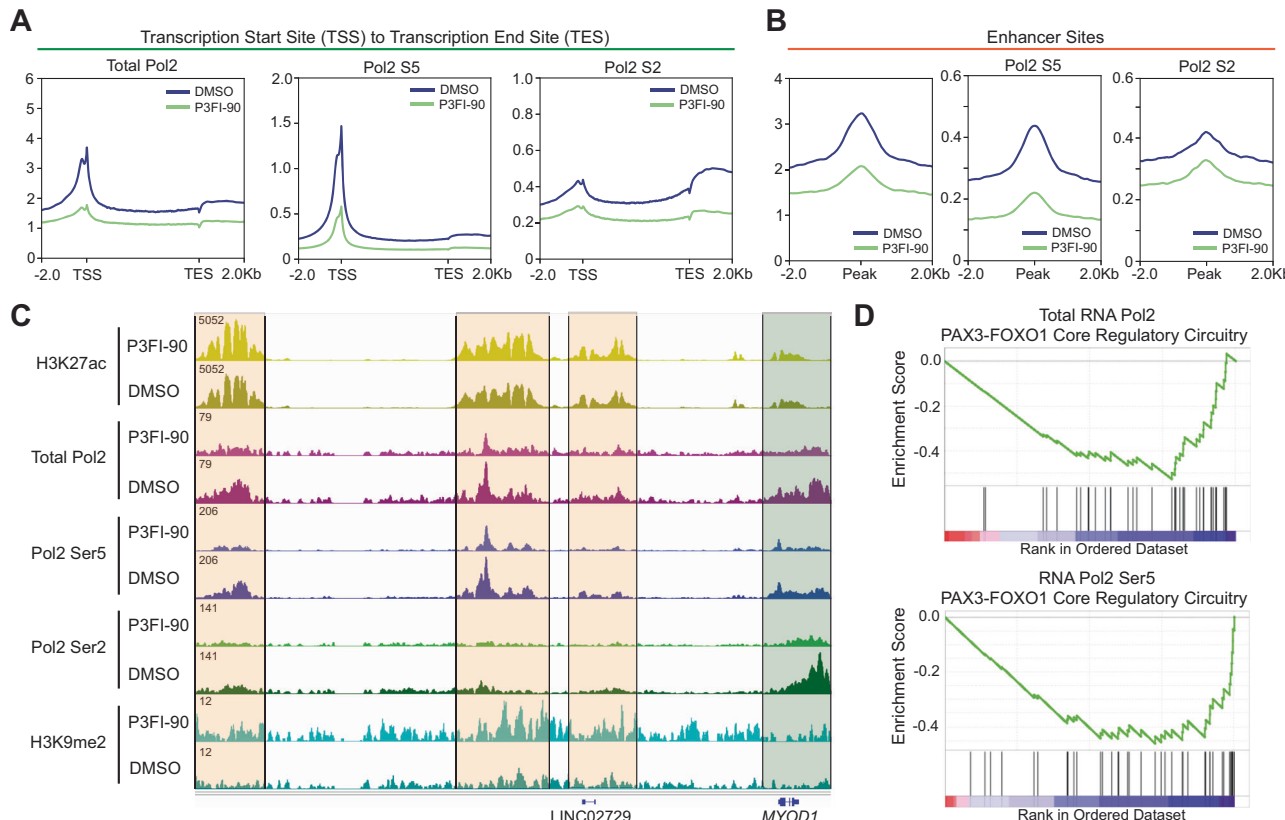

**Fig. 6 | RNA Pol2 analysis of P3FI-90. A** ChIP-seq of Total RNA Pol2, Pol2 Ser5, and Pol2 Ser2 from TSS to transcription end site (TES) genome-wide. $n = 1$. **B** ChIP-seq of Total RNA Pol2, Pol2 Ser5, and Pol2 Ser2 at enhancer sites. $n = 1$. **C** IGV track view of *MYOD1* gene. Orange highlight for enhancer regions and green highlight for gene body. **D** GSEA of ranked differential peaks of RNA Pol2 after treatment with P3FI-90 vs DMSO showing negative enrichment of PAX3-FOXO1 core regulatory circuitry.

## P3FI-90 decreases chromatin loop strength and decreases topologically associating domains (TADs) at PAX3-FOXO1 sites

Given that there were no changes to PAX3-FOXO1, H3K27me3, H3K27ac binding and respective super-enhancer calls following treatment with P3FI-90, we asked if the increase in H3K9me2, HP1, and H3K4me3 is sufficient to change the chromatin structure in FP-RMS cells. We performed Hi-C and assessed if P3FI-90 treatment was associated with decreased number of loops given that H3K9me2 is a heterochromatin mark. We found that there were minimal changes in the number of loops genome-wide (675 loss and 582 gain). Genes associated with these loops did not show any statistically significant association with gene sets. However, we found that although there were minimal differences in loop number, there was decrease in loop strength by aggregate peak analysis (Supplementary Fig. 12A, B). Next, we looked to see if TADs were changed by P3FI-90 given the decrease in loop strength. Interestingly, there was a significant decrease in TAD score in P3FI-90 treated cells compared to DMSO (Supplementary Fig. 12C). Genes associated with the decreased TADs and loop strength were enriched for PAX3-FOXO1 target genes by GSEA (Supplementary Data 5, Table 5). Decrease in TADs and loop strength at PAX3-FOXO1 sites suggests that decrease in PAX3-FOXO1 target transcripts, observed by RNA-seq, may be related to the increase in H3K9me2 and HP1 at PAX3-FOXO1 sites. Additional analysis by HiCRep, CHESS, and Zebra did not reveal significant differences between P3FI-90 and DMSO[28–30].

## P3FI-90 becreased RNA Pol2 binding and phosphorylation

RNA Pol2 binding and phosphorylation of Ser5 results in transcription initiation and phosphorylation of Ser2 leads to elongation. We wanted to determine if P3FI-90 treatment and the resulting increase in H3K9me2 was associated with disruption of RNA Pol2.

We performed ChIP-seq analysis and found that P3FI-90 treatment resulted in a marked decrease in RNA Pol2 binding at TSS and phosphorylation of RNA Pol2 at both Ser5 and Ser2 (Fig. 6A). Furthermore, we saw a decrease in total and phosphorylated enhancer-bound RNA Pol2 (Fig. 6B). This is exemplified by *MYOD1* where there was a significant decrease in RNA Pol2 binding and phosphorylation both at the gene body and enhancers which was associated with increased H3K9me2 (Fig. 6C). Next, we determined the differential RNA Pol2 peaks between P3FI-90 and DMSO and ranked the associated genes. GSEA analysis showed that PAX3-FOXO1 gene sets, and cell cycle genes sets were enriched in downregulated peaks for total Pol2 and Pol2 Ser5. This was consistent with RNA-seq findings of downregulated PAX3-FOXO1 target transcripts and cell cycle arrest demonstrated by flow cytometry (Fig. 6D; Supplementary Data 5, Tables 6–7).

## P3FI-90 inhibits FP-RMS tumor growth in vivo

Finally, we tested P3FI-90 in two xenograft models of FP-RMS using the RH4 cell line. In a metastatic intravenous mouse model, P3FI-90 was able to significantly delay tumor progression measured by luciferase signal (Fig. 7A, $p = 0.0016$). We also tested P3FI-90 in an orthotopic model of FP-RMS injecting tumor cells intramuscularly into the gastrocnemius muscle. We again saw a significant delay in tumor progression measured by tumor volume (Fig. 7B, $p = 0.0046$). There was no significant change in weight during P3FI-90 treatment (Supplementary Fig. 13). We also performed an additional in vivo study using a subcutaneous tumor model. After 5 days of daily treatment with P3FI-90, tumors were collected for RNA-seq analysis. GSEA showed downregulation of PAX3-FOXO1 nuclear targets, and enrichment for myogenesis and apoptosis gene sets after P3FI-90 treatment versus DMSO, consistent with our in vitro findings (Fig. 7C; Supplementary Data 6, Table 1–3).

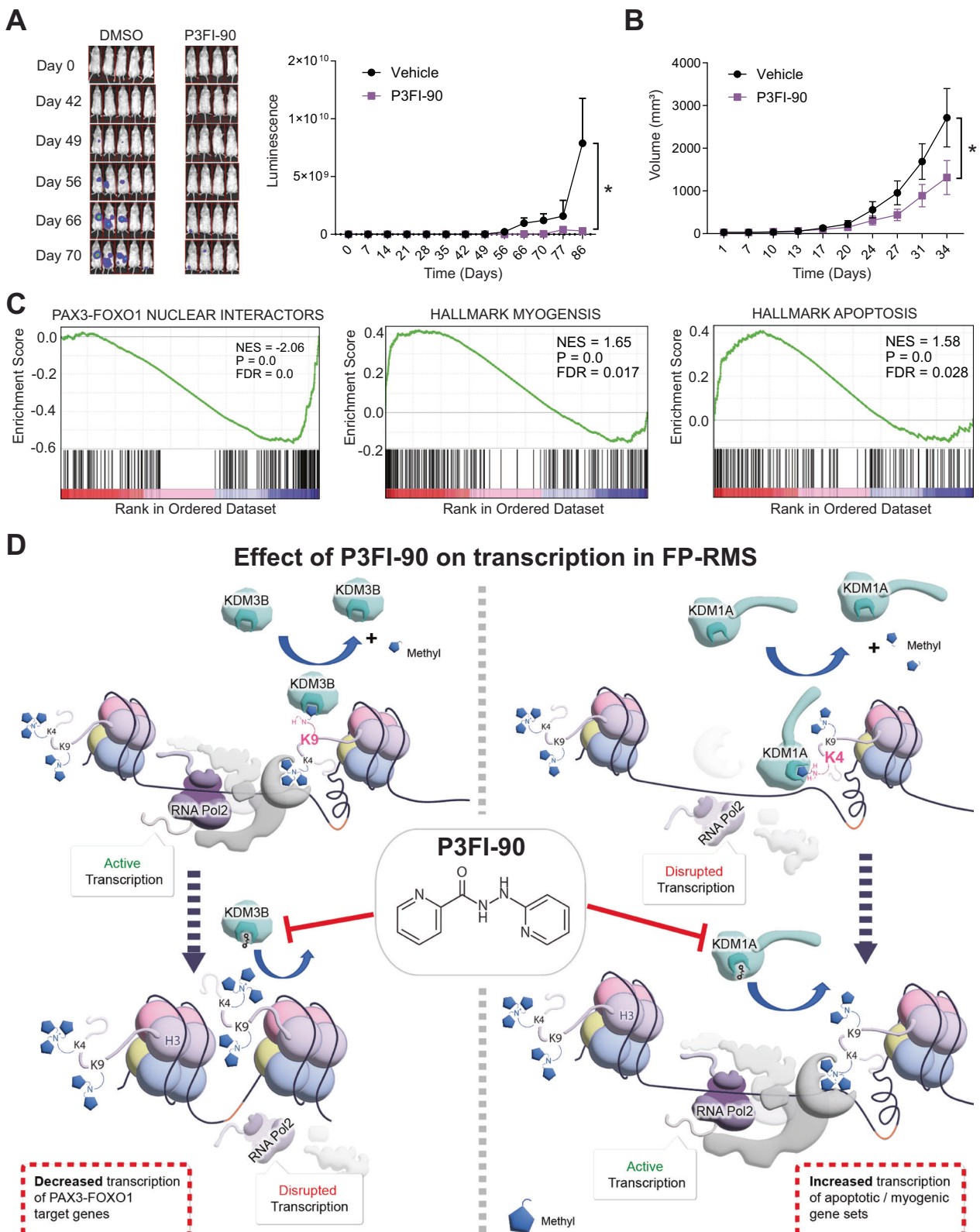

**Fig. 7 | Efficacy of P3FI-90 in metastatic and orthotopic models FP-RMS and pictorial depiction of P3FI-90 action in FP-RMS. A** In vivo xenograft model of metastatic FP-RMS showing delay in tumor progression when treated with P3FI-90 vs DMSO. Data presented as mean values ± SEM. $n = 5$ for DMSO and $n = 4$ for P3FI-90. *$p = 0.0016$ (2-way ANOVA). Error bar = Standard error. **B** In vivo xenograft model of orthotopic intramuscular FP-RMS. $n = 4$ and *$p = 0.0046$ (2-way ANOVA). Error bar = Standard error. **C** RNA-seq GSEA of a tumor isolated from in vivo subcutaneous xenograft model of FP-RMS after treatment with P3FI-90 vs DMSO. **D** Model of P3FI-90 action in FP-RMS. Left panel show active transcription of PAX3-FOXO1 target genes which is downregulated by inhibition of KDM3B by P3FI-90 resulting in increased methylation at H3K9. Right panel show disrupted transcription of apoptosis and myogenesis genes which becomes active due to inhibition of KDM1A by P3FI-90 resulting in increased methylation at H3K4. Source data are provided as a Source Data file.

## Discussion

We show in this study that the compounds, P3FI-63 and P3FI-90, inhibit multiple KDMs with highest selectivity for KDM3B. We also show that FP-RMS are sensitive to KDM3B inhibition by P3FI-63 and P3FI-90. P3FI-63 and P3FI-90 downregulated PAX3-FOXO1 target transcripts without changing PAX3-FOXO1 protein or PAX3-FOXO1 binding levels at enhancers. Interestingly, despite H3K9me2 being a repressive mark and significantly increased after P3FI-90 treatment, deposition of the activating H3K27ac mark was not affected indicating that enhancer/super-enhancer designations were unchanged. Moreover, increase in H3K9me2 and associated increase in HP1 after P3FI-90 treatment did not change the number of loops as expected given that they are markers of heterochromatin. However, we did see a decrease in loop strength and TADs at PAX3-FOXO1 sites. These findings suggests a mechanism whereby inhibition of KDM3B and the resulting increase in H3K9me2 can act as a dial to dampen transcription while maintaining super-enhancer loops and binding of oncogenic transcription factors. In line with this observation, P3FI-90 treatment resulted in decreased RNA Pol2 binding and phosphorylation.

Discovery of KDM3B as a potential target for therapy in FP-RMS extends the findings with other KDM inhibitors reported to be effective in RMS, including the KDM1A inhibitor GSK690, KDM4A inhibitor QC6352, and the pan-KDM inhibitor JIB-04[31–33]. To date, only one other group has reported the identification of potential KDM3B inhibitors (JDI-4 and JDI-12)[34]. However, the ability of JDI-4 and JDI-12 to inhibit other KDMs has not been tested; considering the high homology between KDM family members, it is difficult to conclude KDM3 specificity without direct comparison.

We also observed an increase in H3K4me3 due to P3FI-90 inhibition of KDM1A and KDM5A. Interestingly, although both KDM1A and KDM5A target H3K4 methylation, only KDM1A knockdown showed enrichment of apoptosis gene sets. Negative enrichment of some PAX3-FOXO1 target gene sets was also observed when KDM1A was knocked down (Fig. 3D), consistent with the fact that KDM1A can demethylate H3K9 in certain contexts[35]. These results demonstrated that KDM1A is also a relevant target of P3FI-90's biological action.

We propose a theoretical model of P3FI-90 action based on our findings which requires confirmatory studies but helps to create a framework for future discussion. We propose that P3FI-90 downregulates PAX3-FOXO1 target gene expression by increasing H3K9me2 mainly through the inhibition of KDM3B with potential contribution from KDM1A inhibition (Fig. 7D). Given that FP-RMS is "addicted" to the PAX3-FOXO1-driven super-enhancer circuitry, it seems likely that this circuitry is disproportionately sensitive to increases in H3K9me2 marks leading to decreases in RNA Pol2 at PAX3-FOXO1 sites. P3FI-90 activation of the myogenic differentiation program and induction of apoptosis is likely via increases in H3K4 methylation predominantly by inhibition of KDM1A (Fig. 6D). Overall, H3K9me2 and H3K4me3 increase leads to transcriptional rewiring resulting in sensitivity of FP-RMS to P3FI-90. In this regard, due to the functional redundancy of KDM family members for H3K substrates, simultaneously targeting several KDMs is expected to show increased efficacy than targeting individual KDMs[32]. As KDM3B has been implicated in taxane-platin-resistant small-cell lung cancer, acute lymphoblastic leukemia, and hepatocellular carcinoma, P3FI-90 emerges as a lead compound for further development using structure-activity relationship (SAR) to identify clinical grade KDM3B inhibitors against multiple cancers[36–38]. Also, because P3FI-90 exerted potent growth inhibitory activity on Ewing's sarcoma and osteosarcoma cell lines (Supplementary Fig. 1D), KDM3B inhibition could be effective in these pediatric malignancies. Although in vitro EC50 showed that primary human fibroblasts had similar sensitivity to P3FI compounds, P3FI-90 was well tolerated in mouse models of RMS without significant weight loss at 25 mg/kg indicating that there is likely a therapeutic window. In conclusion, P3FI-90 inhibits multiple KDMs with highest selectivity for KDM3B that can be further developed for the therapy of highly malignant FP-RMS and possibly other "transcriptionally addicted" cancers.

## Methods

All research conducted for this manuscript complies with ethical regulations including approval by the National Institute of Health's Institutional Animal Care and Use Committee, protocol PB-057.

### Cell lines

Cell lines were tested for mycoplasma and identities were ensured by RNA-seq and genotyping. RMS cell lines RH4 and RH30 were gifts from Dr. Peter Houghton, SCMC from Dr. Janet Shipley, RD, and CTR from Dr. Lee Helman. Osteosarcoma cell lines OSA and HU09 and Ewing's Sarcoma cell lines TC-32 and A673 were from Dr. Paul Meltzer. Human fibroblast cell line 7250 (CRL-7250) was obtained from ATCC. All cell lines except OSA and HU09 were grown in DMEM (Quality Biological, #112-013-101CS) supplemented with 10% FBS (ThermoFisher, #A31605-02), 2 mM L-glutamine (Quality Biological, #118-288-061), and 100 U/ml penicillin/streptomycin (ThermoFisher, #15140122). OSA and HU09 were grown in RPMI (Quality Biological, #112-024-101CS) with the same supplements as DEME.

### Small molecules

High throughput screening of the 62,643 compounds from the NCI Molecular Targets Program was carried out in 384-well plates as previously described[5]. P3FI-63 (PubChem CID 912764) and P3FI-90 (PubChem CID 12499590) were purchased from Enamine (EN300-6731773, EN300-262365 respectively).

### In vitro dose-response

Dose responses were performed by quantifying percent cell confluence from phase contrast images using Incucyte ZOOM in 384-well plate format. Dose-response was achieved using a range between 30 μM to 0.17 nM in triplicates. $EC_{50}$ values were calculated for each time point using GraphPad 8 using nonlinear regression variable slope four-parameter function with default setting.

### In vitro enzyme inhibition assay

Enzyme inhibition assay for KDM2B, KDM3A, KDM3B, KDM4B, KDM5A, KDM5B, KDM6A, KDM6B, HDAC1, HDAC2, HDAC3, PRMT5 were performed by Eurofins (St Charles, MO) (Supplementary Methods) KDM1 inhibition was performed using the LSD1 Activity Quantification Assay Kit (Abcam, ab113460) per default protocol (Supplementary Methods). $IC_{50}$ values were calculated using GraphPad 8 nonlinear regression variable slope four-parameter function with default setting.

### Molecular docking

Compounds have been docked into KDMs' x-ray structures using ICM-Pro software (Molsoft LLC, San Diego, CA). The PoketFinder software (Molsoft) was used for the identification of the pockets for docking. Docking was performed under the conditions of flexible receptor side chains. The following structures from the Protein Data Bank (PDB) have been used in docking: PDB: 4C8D, 5R7X, 5RAA, 5RAM, 5RAN, 5RAO, 5RAZ and 6RBJ for KDM3B; PDB: 7JM5 for KDM4B; PDB: 6DQ40 for KDM5A; PDB: 5A3T for KDM5B; PDB: 6G8F for KDM6A; PDB: 5FP3 and 4ASK for KDM6B; and PDB: 2Z3Y for LSD1. ICM-Pro was used for interaction analysis, Radial Convolutional Neural Net (RTCNN) scores, and generation of images of the predicted complexes of P3FI-90 with KDMs.

### RNA-seq

RNA-seq was performed as previously described[39]. Briefly, total RNA collected using RNeasy Plus Mini kit (Qiagen, 74134). Poly-A selected RNA libraries were prepared and sequenced on an Illumina

NextSeq500 to a depth of 10 million reads per sample. Reads were aligned to hg19 using STAR version 2.5.3a, and gene expression was calculated as Transcripts per Million mapped reads (TPM) using RSEM version 1.3.1 and the UCSC reference. For variance normalization, DESeq2 package was utilized. GSEA analysis was performed on log2-fold change ranked list calculated by comparison with control in samples with single experiments using Run GSEAPreRanked with default settings (Detailed gene sets can be found in Supplementary Data 7, Table 3). For samples with biological triplicates, Run GSEA function was used with default settings except for Permutation Type, for which Gene_Set was used and Collapse/Remap to gene symbols for which No Collapse was chosen. Volcano plots of the triplicate samples can be found in Supplementary Fig 14A, B.

### scRNA-seq
RH4 cells were treated with P3FI-90 at 1 μM for 16 h and collected. 10X Genomics platform was used for scRNA-seq per protocol. Library was sequenced on Illumina NextSeq2000 to 300 million reads per sample. Fastq files were aligned using cellranger (10X Genomics) V6 using hg19 with default settings. Barcodes filtration was performed based off the distribution of data (median ± 3 deviations) for high percentage of mitochondria, low number of genes, and low number of cells. We used the standard cutoff of at least 200 features per cell. Percent mitochondrial read range was <15% for each cluster (Supplementary Fig. 10B-C). Individual sample normalization was done using SCTransform through Seurat V3 setting the variable.features.rv.th to 1.3[40]. No other factors including MT reads were used for normalizations. Doublets were detected using DoubletFinder V2 with a doublet estimate of 3% and removed before clustering. Cell cycle annotations was determined using the CellCycleScoring method in Seurat. Sample integration, clustering, dimension reduction, and differential expression was performed using the Seurat V3 Package. Data integration was performed using the default parameters followed by clustering using the Leiden algorithm[41] and a resolution of 0.2 and projected using Uniform Manifold Approximation and Projection (UMAP)[42]. Differential expression was performed using the FindMarkers function in Seurat with the option of using MAST[43] for method which uses a hurdle model for handling the zero-inflated single-cell data. GSEA analysis was performed by using the fold change from the differential expression analysis and fed into fGSEA[44]. For the heatmap generation, the leading edge genes from the GSEA leading edge analysis from Cluster 2 PAX3-FOXO1 signature (Gryder_PAX3-FOXO1_CRC Gene Set) and the Myoblast and Myocyte signatures (Supplementary Data 4, Tables 1–2) were utilized. The code utilized for scRNA-seq analysis is available on GitHub at https://github.com/CCRGeneticsBranch/scRNAseq_snakemake_pipeline/tree/main.

### ChIP-seq
ChIP-seq was performed as previously described[45]. Cells were fixed using formaldehyde (1% final concentration) for 14 min and sheared for 27 cycles 30 s on/30 s off (Active Motif, 53052). Equal amount (50 ng) of Drosophila spike-in (Active Motif, 53083) was added to the chromatin when setting up the immunoprecipitation. Samples were immunoprecipitated overnight at 4 °C with antibodies (including 2 μg of spike-in antibody) and purified by agarose beads (Active Motif, 53040). Please refer to antibody table for details including the in-house anti-PAX3-FOXO1 antibody (Supplementary Data 7, Table 1)[46]. Also, please refer to Supplementary Data 7, Table 2 for details on replicates, read depth, peaking, and p-values. ChIP library was made using TruSeq ChIP library prep kit (Illumina, IP-202-1012). Libraries were sequenced on Illumina NextSeq500. ChIP-seq reads were aligned to hg19 using BWA version 0.7.17. Samples were normalized using a scaling factor Drosophila spike-in reads per million mapped dm3. Peaks called using MACS2 (version 2.1.1.20160309, https://github.com/taoliu/MACS) using "narrow" mode for all targets except for H3K27me3

and H3K9me2 for which "broad" was used. Mapping artifacts were removed (locations black-listed by ENCODE, https://sites.google.com/site/anshulkundaje/projects/blacklists). The spike-in normalized read count for merged peaksets was calculated using deepTools' "multiBigwigSummary"[47]. The nearest genes were selected with bedtools' "closest"[48]. For comparisons with single samples, pre-ranked log2-fold changes were calculated and GSEA performed by "gsea-cli.sh GSEAPreranked -scoring_scheme weighted" command with FDR cutoff 0.25. For samples with triplicates, we used DiffBind's dba.count function to calculate the read count of peak sets[49]. The count matrix was normalized by spike-in reads. Then, we used DESeq2 to perform differential peak analysis with adjusted p-value cutoff 0.05 and fold change cutoff 1.5.

### ChromHMM
The chromatin states were characterized using chromHMM[50] which can learn and infer the combinatorial patterns of histone marks using Hidden Markov Model algorithm. The read count for each state was calculate with deepTools' "multiBamSummary" command[47]. The percentage of genome coverage for each state was calculated with in-house bash script. ChIP-seq utilized to define states and the ChromHMM methods were previously published[6].

### ATAC-seq
ATAC-seq was performed as previously described[24]. Cells were treated with DMSO or P3FI-63 and collected. 50,000 cells were counted and nuclei isolated. Tagmentation was performed (Illumina, 20034198) and purified using MinElute kit (Qiagen, 28206). Library was amplified (NEB, M0541L) for 16 cycles. Library was purified using AMPure XP beads (Beckman, A63880) and sequenced using NextSeq500. Peaks were called using MACS2 (version 2.1.1.20160309, https://github.com/taoliu/MACS) using "narrow" mode. Normalized read count for promoter regions of each gene was calculated using deepTools' "multiBigwigSummary". To identify the nearest genes, we used annotatePeaks.pl from HOMER suite to annotate the peaks[51]. Then, we calculated the log2-fold changes for the nearest peak for each gene to generate the GSEA rank file. Finally, we used GSEA's[8] "gsea-cli.sh GSEAPreranked -scoring_scheme weighted" command on ranked gene list to generate gene set enrichment list with FDR cutoff 0.25. QC profile of ATAC-seq using plotHeatmap from deepTools showing enrichment of peaks at promotor region can be found in Supplementary Fig. 14C.

### Hi-C
RH4 cells were treated with P3FI-90 at 1 μM for 24 h and collected. The Dovetail Micro-C kit (Dovetail, 21006) was used to generate libraries. In short, MNase is used to digest the chromatin after fixation. Proximity ligation is performed and after reverse crosslinking, library is synthesized and purified using streptavidin-based bead purification. 3 separate libraries of technical replicates were sequenced to the total depth of 800 million reads. The raw reads were mapped to hg19 genome using a bash script omni-c_qc.sh from Dovetail Github (https://github.com/dovetail-genomics). The replicates were then merged with Pairtools' "merge"[52]. The active and inactive compartments were assigned by FAN-C[53] with "fanc compartments" at 500 kb resolution. Eigenvectors was corrected by Spearman correlation with H3K4me1. Then we looked for the flipped compartments with a cutoff of loading difference 0.02. The flipped compartments were intersected with known coding genes (GRCh37/hg19 RefSeq annotation) ≥1 TPM in either DMSO or P3FI-90 RNA-seq. Enrichment analysis was performed by overlapping the flipped gene list with NCI v30 gene sets using R package "GeneOverlap"[54]. The p-values were adjusted using R method "p.adjust". For loop calling, we used the FAN-C command "fanc loops" with 10 kb resolution. Loops were further filtered with "rh-filter" and "remove-singlets" options. The lost and gained loops were determined

using bedtools[48] command "pairToPair". To identify the loop associated genes, we used "pairToBed" command in bedtools. For TAD analysis, we first used HiCExplorer's[55] hicFindTADs command with FDR cutoff 0.01 at 5 kb resolution. Then we compared DMSO vs P3FI-90 by using HiCExplorer's "hicDifferentialTAD" command. We further filtered the rejected TAD regions by adjusting intra-TAD $p$-values with cutoff 0.05. Then called loops in differential TADs were intersected with transcription start site regions of the coding genes. GSEA performed by overlapping with NCI v30 gene sets using R package "GeneOverlap". We performed Aggregate Peak Analysis on loops of interests using Juicer's "apa" command[56]. The contact matrices were normalized using the interactions identified in spike-in mouse reads.

## Tiled KDM3B CRISPR knockout screen

sgRNA guides were designed throughout all KDM3B exons resulting in 479 sgRNA and cloned into lentiCRISPR v2 (gift from Dr. Feng Zhang, Addgene plasmid # 52961; http://n2t.net/addgene:52961; RRID:Addgene_52961). Lentivirus library was produced by transfecting HEK293 cells (Thermo, L3000015) and virus collected at 24, 48, and 72 h and pooled. PEG-it (SBI, LV810A-1) was used to concentrate the virus. RH4 cells were transduced at MOI of 0.2 and Puromycin (Thermo, A1113803) selected for 7 days. Cells were collected every 5 days for sequencing. DNA was isolated using DNA isolation kit (Qiagen, 80204). gRNA inserts were amplified using primers harboring Illumina TruSeq adapters with i5 and i7 barcodes, and the resulting libraries were sequenced on NextSeq500[57]. MAGeCK was used for pairwise comparison to the plasmid pool[58]. Log fold depletion of guides were calculated and walking average of 4 guides were plotted for the entire KDM3B gene.

## Western blot analysis

Whole protein cell lysates were obtained by using RIPA buffer while total histone cell extracts were obtained by using EpiQuick Total Histone Extraction Kit (Epigentek, OP-0006). Protein concentrations were estimated by Pierce BCA Protein Assay Kit (Thermo, 23225). Refer to antibody table for details (Supplementary Data 7, Table 1). Membranes were developed using ECL reagent (GE Healthcare, RPN2106).

## Apoptosis quantification

RH4 and RH30 cells were treated with P3FI-63 and P3FI-90 for 24 h. Caspase-Glo 3/7 kit (Promega, G8091) was used, and luminescence measured for quantitation of apoptosis.

## Flow cytometry

SCMC cells were treated with P3FI-90 at 1 μM for 24 h vs DMSO. Cells were fixed and permeabilized before stained using antibody against Ki67 and pH3 followed by secondary Alexa488 antibody and Hoechst (Supplementary Data 7, Table 1). Cells were analyzed on BD FAC-Symphony A5. Gating strategy shown on Supplementary Fig. 10D.

## KDM3B protein purification

Enzymatic domain of KDM3B for the NMR experiments were a generous donation from Dr. Paul Brennan through the Structural Genomics Consortium as glycerol stock of E coli with KDM3B expression plasmid (https://doi.org/10.5281/zenodo.1219696). The plasmid expression vector JMJD1BA-c068 used was a gift from Dr. Nicola Burgess-Brown (Addgene plasmid #53627; http://n2t.net/addgene:53627; RRID:Addgene_53627). Protein was expressed in *E. Coli* and purified using His trap column (Cytiva, #17531901) followed by sephacryl S200 SEC (Cytiva, #17058410) column. The full-length KDM3B was synthesized by GeneUniversal (Newark, DE) with an upstream hexahistidine tag and Avi tag (His6-Avi-KDM3B) and formatted for Gateway cloning. An entry clone was generated and subcloned into baculovirus destination vector pDest-008 (Thermo Fisher

Scientific, #11804010). Bacmid DNA encoding His6-Avi-KDM3B was generated using the Bac-to-Bac system (Thermo Fisher Scientific, #10359016) using the manufacturer's instructions. Spodoptera frugiperda (Sf9) cells were transfected with bacmid DNA using Cellfectin (Thermo Fisher Scientific, #10362100). Harvested baculovirus was tittered and Trichoplusia ni (TniFNL) cells were infected at an MOI of 3. HisTrap FF column (Cytiva, #17531901) was used to purify the KDM3B protein and in vitro biotinylation of the protein was performed with components mixed in the following molar ratio: 7.5 parts KDM3B, 1 part biotin ligase (untagged), 1250 parts ATP, 75 parts biotin. This was followed by another HisTrap FF purification. Full-length KDM3B was used in the SPR experiment.

## KDM3B enzyme activity assay

Full-length KDM3B was tested for enzymatic function using histone tail with K9 dimethylation (EpigenTek, R-1026-200). Enzyme activity was performed in 50 mM HEPES pH 7.5, 1 mM TCEP, 100 μM 2-OG, 100 μM Ascorbate, and 50 μM Iron Sulfate for 3 h at room temp. Histone demethylation reactions were monitored by mass spectrometry measuring the mass of the histone H3 K9me2 substrate. Peptide masses were analyzed on a X500B Q-TOF (SCIEX) mass spectrometer coupled to an Exion UHPLC. Reactions were lyophilized before analysis and resuspended in 0.1% formic acid before injection onto an Synergi Fusion-RP C18 column (Phenomenex). Data were analyzed using Explorer and mass reconstruction was performed in Bio ToolKit, both within the SCIEX OS software package.

## NMR

NMR spectra were recorded on Bruker AVANCE III 500 MHz spectrometer equipped with a Bruker TCI cryogenically cooled probe. Samples were prepared in 90% aqueous buffer containing 10% $D_2O$ for sample locking. Given the difficulty maintaining iron oxidative state as previously described, iron was replaced with zinc using the buffer as per Leung et al.[59], and composed of 50 mM Tris-d11, pH 7.5, and 80 μM zinc sulfate monohydrate. P3FI-90 was at 300 μM and KDM3B at 15 μM. Suppression of the water peak was through excitation sculpting for the WaterLOGSY experiments using the pulse program ephogsyno.2 or through presaturation for the CPMG experiments (pulse program cpmgpr1d). A reference 1H spectrum was collected for P3FI-90 peaks and annotated based on supporting [13]C and HSQC spectra. For WaterLOGSY (WL) experiments, spectra were acquired with 256-512 scans[60], while for CPMG experiments, spectra were acquired with 64 scans.

## SPR

Full-length KDM3B and P3FI-90 were assessed using Biacore T200 system (GE Healthcare). Neutravidin was covalently immobilized on a CM5 chip. Avi-tagged KDM3B protein was added to FC2 to a final response of ~800 RU. P3FI-90 was prepared in degassed, filtered HBS-P + (GE Healthcare) buffer with 3% DMSO. Single cycle kinetic experiments were performed using five injections (30 μL min⁻¹) of increasing concentrations (20–2000 nM) passed over the sensor chip for 150-s association, followed by a 420-s dissociation.

## CRISPRi knockdown of KDMs

Plasmid construct for lentivirus containing dCas9-KRAB (gift from Dr. Charles Gersbach, Addgene plasmid #71236; http://n2t.net/addgene:71236; RRID:Addgene_71236) with target sgRNA was synthesized against KDM1A, KDM3B, KDM4B, and KDM5A. sgRNA sequences are as follows. KDM1A: caccgccacggagcgacagagcgag. KDM3B: caccgcagcggcgagcggagatccg KDM4B: caccgcggtcgcca gcaaccgagcg. KDM5A: caccgcgtctttgagccgagttggg. Lentivirus was produced by transfecting HEK293 cells and pooling virus collection at 24, 48, and 72 h. CRISPRi knockdown cells were selected using Puromycin for 7 days and cells collected for analysis.

## In vivo xenograft model of FR-RMS

All animal experiments were approved by Institutional Animal Care and Use Committee. NSG mice are housed under IVM Microisolator conditions with light cycle 6 am to 6 pm. Maximum size of tumor allowed through our IACUC is one dimension no greater than 1.8 cm at which point mice must be euthanized. This limit has not been exceeded in this study. Additional moribund status and weight loss > 10% are conditions for euthanasia. No mice met the condition for euthanasia due to moribund status or weight loss. Anesthesia was performed using Somni Scientific vaporizer set at 2% isoflurane for IVIS imaging. In the orthotopic IM model, 7-week-old female NSG mice were injected with 5 million RH4 cells with Matrigel into the gastrocnemius muscle. Treatment with P3FI-90 started 5 days later at 20 mg/kg body weight dosed daily i.p. for 5 days with 2-day rest. Dose was well tolerated so P3FI-90 was increased to 25 mg/kg on day 28 of treatment until the end of study. In the IV metastatic model, 12-week-old female NSG mice were injected with 1 million RH4 cells expressing luciferase. Treatment with P3FI-90 was started 3 days later at 25 mg/kg body weight given daily i.p. for 5 days with 2 days rest. Only female mice were used in this study which allows for housing of 5 mice per cage. Sex consideration was based on practical feasibility issues. This included aggressiveness of male mice with propensity to injure one another which is especially concerning in severely immunocompromised NSG mice. For tumor analysis, 5 million RH4 cells were injected subcutaneously. P3FI-90 treatment started after 12 days at 25 mg/kg dose i.p. daily for 5 days at which time tumors were collected and snap frozen. RNA was processed with RNA-seq as above. Mouse euthanasia is performed using carbon dioxide at a flow rate of 5.5 liters per minute until breathing ceases plus additional 2 more minutes. Mouse is removed from carbon dioxide-filled cage and observed for movement for 60 s to insure death.

## Statistical analysis

Statistical analysis performed for bulk RNA-seq, scRNA-seq, ChIP-seq, and HiC are described in their respective sections above. GraphPad 8.4.3 was used for $EC_{50}$ and $IC_{50}$ calculations (four-parameter non-linear regression), caspase quantitation and flow cytometry (t-test), and in vivo mouse studies, (analysis of variance).

## In vitro Enzyme Inhibition Assay

Below are Eurofins protocols for the Enzyme Inhibition Assay.

**KDM2B.** Activity of the human KDM2B is quantified by TR-FRET. After the demethylase activity on the biotinylated substrate, the demethylated peptide is captured by the specific Eu-labeled antibody (Eu-donor or Eu-Ab). The test compound, reference compound or water (control) is mixed with about 3 ng of enzyme in a buffer containing 20 mM MOPS/Tris (pH 7), 5 µM FAS, 100 µM ascorbic acid, 1 µM 2-oxoglutarate, 0.01% Tween 20 and 0.01% BSA. Thereafter, the reaction is initiated by adding 24 nM of the biotin-H3K36me2 substrate (H-RKAAPATGGVK(Me2)KPHRYRPGTVK(Biotin)-OH, Anaspec #AS64601), and the mixture is incubated for 10 min at room temperature. For basal control measurements, the enzyme is omitted from the reaction mixture. Following incubation the reaction is stopped by adding 1 mM EDTA. After 5 min, the anti-methyl histone H3K36me1 antibody is added. After 10 min the anti rabbit IgG antibody labeled with europium chelate is added. After 10 min the Ulight streptavidine is added. After 60 min more, the fluorescence transfer is measured at λex = 320 nm and λem = 620 nm and λem = 665 nm using a microplate reader (Envision, Perkin Elmer). The enzyme activity is determined by dividing the signal measured at λem = 665 nm by that measured at 620 nm (ratio). The results are expressed as a percent inhibition of the control enzyme activity. The standard inhibitory reference compound is 2,4 PDCA, which is tested in each experiment at several concentrations to obtain an inhibition curve from which its IC50 value is calculated.

Reference: ROTILI, D. and MAI, A. (2011), Targeting Histone Demethylases: A New Avenue for the Fight against Cancers, Genes & Cancer, 2: 663.

**KDM3A.** Activity of the human JMJD1A is quantified by TR-FRET. After the demethylase activity on the biotinylated substrate, the demethylated peptide is captured by the specific Eu-labeled antibody (Eu-donor or Eu-Ab). The test compound, reference compound or water (control) is mixed with about 5 ng of enzyme in a buffer containing 50 mM Hepes/Tris (pH 7.5), 5 µM FAS, 100 µM ascorbic acid, 0.01% Tween 20 and 0.01% BSA. Thereafter, the reaction is initiated by adding 100 nM of the biotin-H3K9me1 substrate (H-ARTKQTARK(Me)STGGKAPRKQ-LAGGK(Biotin)-OH, Anaspec, #AS64358) and 1 µM 2-oxoglutarate, then the mixture is incubated for 5 min at room temperature. For basal control measurements, the enzyme is omitted from the reaction mixture. Following incubation the reaction is stopped by adding 1 mM EDTA. After 5 min, the anti histone H3K9 antibody labeled with europium chelate and the Ulight streptavidine are added. After 60 min more, the fluorescence transfer is measured at λex = 320 nm and λem = 620 and λem = 665 nm using a microplate reader (Envision, Perkin Elmer). The enzyme activity is determined by dividing the signal measured at λem = 665 nm by that measured at 620 nm (ratio). The results are expressed as a percent inhibition of the control enzyme activity. The standard inhibitory reference compound is 2.4 PDCA, which is tested in each experiment at several concentrations to obtain an inhibition curve from which its IC50 value is calculated.

Reference: Heightman T. D. (2011), Chemical biology of lysine demethylases, CurrenChemical Genomics, 5: 62.

**KDM3B.** Activity of the human KDM3B is quantified by TR-FRET. After the demethylase activity on the biotinylated substrate, the demethylated peptide is captured by the specific Eu-labeled antibody (Eu-donor or Eu-Ab). The test compound, reference compound or water (control) is mixed with about 1.5 ng of enzyme in a buffer containing 45 mM Hepes/Tris (pH 7.5), 5 µM FAS, 100 µM ascorbic acid, 10 µM 2-oxoglutarate, 0.01% Tween 20 and 0.01% BSA. Thereafter, the reaction is initiated by adding 60 nM of the biotin-H3K9me1 substrate (H-ARTKQTARK(Me)STGGKAPRKQLAGGK(Biotin)-OH, Anaspec, #AS64358), and the mixture is incubated for 10 min at room temperature. For basal control measurements, the enzyme is omitted from the reaction mixture. Following incubation the reaction is stopped by adding 1 mM EDTA. After 5 min, the anti histone H3K9 antibody labeled with europium chelate and the Ulight streptavidine are added. After 60 min more, the fluorescence transfer is measured at λex = 320 nm and λem = 620 and λem = 665 nm using a microplate reader (Envision, Perkin Elmer). The enzyme activity is determined by dividing the signal measured at λem = 665 nm by that measured at 620 nm (ratio) The results are expressed as a percent inhibition of the control enzyme activity. The standard inhibitory reference compound is 2.4 PDCA, which is tested in each experiment at several concentrations to obtain an inhibition curve from which its IC50 value is calculated.

Reference: Kim, J.-Y., Kim K.-B., Hyeon Eom, G., Choe, N., Jin Kee, H., Son, H.-J., Oh, S.-T., Kim, D.-W., Ho Pak, J., Jo Baek, H., Kook, H., Hahn, Y., Kook, H., Chakravarti, D., and Seoa, S.-B. (2012), KDM3B Is the H3K9 Demethylase Involved in Transcriptional Activation of lmo2 in Leukemia, MCB, 32: 2917.

**KDM5A.** Activity of the human KDM5A is quantified by TR-FRET. After the demethylase activity on the biotinylated substrate, the demethylated peptide is captured by the specific Eu-labeled antibody (Eu-donor or Eu-Ab). The test compound, reference compound or water (control) is mixed with about 7.5 ng of enzyme in a buffer containing 45 mM Hepes/Tris (pH 7.5), 5 µM FAS, 100 µM ascorbic acid, 3 µM 2-oxoglutarate, 0.01% Tween 20 and 0.01% BSA.

Thereafter, the reaction is initiated by adding 100 nM of the biotin-H3K4me3 substrate (H-ARTK(Me3)QTARKSTGGKAPRKQLAGGK(Biotin)-NH2, Anaspec #AS64192), and the mixture is incubated for 5 min at room temperature. For basal control measurements, the enzyme is omitted from the reaction mixture. Following incubation the reaction is stopped by adding 1 mM EDTA. After 5 min, the anti-methyl histone H3K4me1-2 antibody labeled with europium chelate and the Ulight streptavidine are added. After 60 min more, the fluorescence transfer is measured at λex = 320 nm and λem = 620 nm and, λem = 665 nm using a microplate reader (Envision, Perkin Elmer). The enzyme activity is determined by dividing the signal measured at λem = 665 nm by that measured at 620 nm (ratio). The results are expressed as a percent inhibition of the control enzyme activity. The standard inhibitory reference compound is 2,4 PDCA, which is tested in each experiment at several concentrations to obtain an inhibition curve from which its IC50 value is calculated.

Reference: NOTTKE, A., COLAIACOVO, M.P. and SHI, Y (2009), Developmental roles of the histone lysine demethylases, Development, 136: 879.

**KDM5B**. Activity of the human KDM5B is quantified by TR-FRET. After the demethylase activity on the biotinylated substrate, the demethylated peptide is captured by the specific Eu-labeled antibody (Eu-donor or Eu-Ab). The test compound, reference compound or water (control) is mixed with about 0.75 ng of enzyme in a buffer containing 45 mM Hepes/Tris (pH 7.5), 5 μM FAS, 100 μM ascorbic acid, 10 μM 2-oxoglutarate, 0.01% Tween 20 and 0.01% BSA. Thereafter, the reaction is initiated by adding 60 nM of the biotin-H3K4me3 substrate (H-ARTK(Me3)QTARKSTGGKAPRKQLAGGK(Biotin)-NH2, Anaspec #AS64192), and the mixture is incubated for 30 min at room temperature. For basal control measurements, the enzyme is omitted from the reaction mixture. Following incubation the reaction is stopped by adding 1 mM EDTA. After 5 min, the anti-methyl histone H3K4me1-2 antibody labeled with europium chelate and the Ulight streptavidine are added. After 60 min more, the fluorescence transfer is measured at λex = 320 nm, λem = 620 nm, and λem = 665 nm using a microplate reader (Envision, Perkin Elmer). The enzyme activity is determined by dividing the signal measured at λnm = 665 nm by that measured at 620 nm (ratio). The results are expressed as a percent inhibition of the control enzyme activity. The standard inhibitory reference compound is 2,4 PDCA, which is tested in each experiment at several concentrations to obtain an inhibition curve from which its IC50 value is calculated.

Reference: Kristensen, L.H., Nielsen, A.L., Helgstrand, C., Lees, M., Cloos, P., Kastrup, J.S., Helin, K., Olsen, L. and Gajhede, M. (2012), Studies of H3K4me3 demethylation by KDM5B/Jarid1B/PLU1 reveals strong substrate recognition in vitro and identifies 2,4-pyridine-dicarboxylic acid as an in vitro and in cell inhibitor, FEBS Journal, 279: 1905.

**KDM6B**. Activity of the human KDM6B is quantified by TR-FRET. After the demethylase activity on the biotinylated substrate, the demethylated peptide is captured by the specific Eu-labeled antibody (Eu-donor or Eu-Ab). The test compound, reference compound or water (control) is mixed with about 200 ng of enzyme in a buffer containing 18 mM Tris/HCl (pH 7.5), 135 mM NaCl, 45 μM FAS, 1.8 mM ascorbic acid, 4.5 μM 2-oxoglutarate and 0.009% Tween 20. Thereafter, the reaction is initiated by adding 200 nM of the biotin-H3K27me3 substrate (H-ATKAARK(Me)SAPATGGVKKPHRYRPGGK(Biotin)-OH, Anaspec, #AS64367), and the mixture is incubated for 10 min at room temperature. For basal control measurements, the enzyme is omitted from the reaction mixture. Following incubation the reaction is stopped by adding 1 mM EDTA. After 5 min, the anti-methyl histone H3K27me1-2 antibody labeled with europium chelate and the Ulight streptavidine are added. After 60 min more, the fluorescence transfer is measured at λex = 320 nm and λem = 620 and λem = 665 nm using a microplate reader (Envision, Perkin Elmer). The enzyme activity is determined by

dividing the signal measured at λem = 665 nm by that measured at 620 nm (ratio). The results are expressed as a percent inhibition of the control enzyme activity. The standard inhibitory reference compound is 2.4 PDCA, which is tested in each experiment at several concentrations to obtain an inhibition curve from which its IC50 value is calculated.

Reference: Rotili, D. and Mai, A. (2011), Targeting histone demethylases: a new avenue for the fight against cancer, Genes Cancer, 2: 663.

**HDAC1, HDAC2, HDAC3**. Activity of the human HDACs is quantified by measuring the formation of fluoro-lysine from a fluorogenic, acetylated peptide substrate for HDAC, using a recombinant enzyme. The test compound, reference compound or water (control) are preincubated for 5 min at room temperature with about 400 ng of enzyme in a buffer containing 45 mM Tris–HCl (pH 8.0), 123.3 mM NaCl, 2.43 mM KCl, 0.9 mM MgCl2 and 0.18% BSA. Thereafter, the reaction is initiated by adding 20 μM of the fluorogenic HDAC substrate. For HDAC1 and HDAC2, substrate protein sequence was Arg-His-Lys-Lys(Ac) (BPS Bio, #50032) and for HDAC3, substrate protein sequence was Gln-Pro-Lys-Lys(Ac) (Enzo Life Sci, #BML-KI179). Fluorescence intensity is measured at λex = 355 nm and λem = 460 nm using a microplate reader (Envision, Perkin Elmer). This measurement at $t = 0$ allows the detection of any compound interference with the fluorimetric detection method at these wavelengths. The mixture is then incubated for 15 min at room temperature. For basal control measurements, the enzyme is omitted from the reaction mixture. After incubation, the reaction is stopped by adding one assay volume of buffer containing 1 cc HDAC developer (peptidase activity, BPS Bioscience) and 20 μM trichostatin A. After 30 min (HDAC2) or 60 min (HDAC1,3), the fluorescence intensity emitted by the reaction product fluoro-lysine is then measured at the same wavelengths ($t = 75$ HDAC1, $t = 45$ HDAC2, $t = 70$ HDAC3). The enzyme activity is determined by subtracting the signal measured at $t = 0$ from that measured at $t = 75, 45, 70$ (HDAC1, HDAC2, HDAC3). The results are expressed as a percent inhibition of the control activity. The standard reference compound is trichostatin A, which is tested in each experiment at several concentrations to obtain a competition curve from which its IC50 is calculated.

Reference: STRAHL, B.D. and ALLIS, C.D. (2000), The language of covalent histone modifications, Nature, 403: 41.

**PRMT5**. Inhibitory effect of compounds on the methyltransferase activity of human PRMT5 complex via the measurement its capacity to transfer methyl groups in a radioactivity assay. The test compound, reference compound, or water (control) are incubated for 120 min at 22 °C with about 200 ng human PRMT5 complex enzyme, 600 nM [3H] SAM and 250 nM Histone H4 full-length (amino acid 2–104 with N-terminal His-tag, Sigma #SRP0178) in a buffer containing 45 mM Tris–HCl (pH 9), 45 mM NaCl, 4.5 mM MgCl2 and 3.6 mM DTT. For control basal measurements, the enzyme is omitted from the reaction mixture. Following incubation, the reaction is stopped by adding 33 mM citric acid, and the samples are filtered rapidly under vacuum through glass fiber filters (GF/B, Packard) presoaked with 33 mM citric acid and rinsed several times with ice-cold 33 mM citric acid using a 96-sample cell harvester (Unifilter, Packard). The filters are dried then counted for radioactivity in a scintillation counter (Topcount, Packard) using a scintillation cocktail (Microscint 0, Packard). The results are expressed as a percent inhibition of the control enzyme activity. The standard reference compound is SAH, which is tested in each experiment at several concentrations to obtain a competition curve from which its IC50 is calculated.

Reference: Yost, J.M., Korboukh, I., Liu, F., Gao, C. and Jin, J. (2011), Targets in epigenetics: inhibiting the methyl writers of the histone code, Curr. Chem. Genomics, 5: 72-84.

**KDM1A/LSD1**. This is an assay by Abcam (Cat# ab113460). This assay is an antibody based detection assay of demethylated product using

fluorescence. Methylated histone substrate (Ala-Arg-Thr-Lys(Me2)-Gln-Thr-Ala-Arg-Lys-Ser-Thr-Gly-Gly-Lys-Ala-Pro-Arg-Lys-Gln-Leu-Ala-Gly-Gly-Lys-Biotin, from Kit), protein nuclear extract (RH4 cells), P3FI-90 (in a dilution series) are mixed in provided Assay Buffer and incubated at 37 C for 120 min. After 3× washes, Capture Antibody (from Kit) is added and incubated at room temp for 60 min. After 3× washes, Detection Antibody is added and incubated at room temp for 30 min. After 4× washes, Fluorescence Development Solution (from Kit) is added and incubated at room temp for 3 min. Fluorescence is read at Ex/Em = 530/590 nM within 10 min. Tecan Plate reader was used to read Fluorescence.

## Reporting summary

Further information on research design is available in the Nature Portfolio Reporting Summary linked to this article.

## Data availability

All sequencing files are available on GEO under the accession number GSE219199. Sequencing files both raw fastq files and processed files included are RNA-seq (TPM matrix text file), ChIP-seq (Peak call bed file), ATAC-seq (peak call bed file), scRNA-seq (barcodes, features, and matrix files), and HiC (merged hic files). Protein crystal structures were accessed through the PDB database. https://www.rcsb.org/. Enricher database was utilized for analysis of ChIP results: https://maayanlab.cloud/Enrichr/[26]. GO database was utilized for analysis of ChIP results: https://geneontology.org/. Majority of the gene sets were obtained from GSEA website: https://www.gsea-msigdb.org/gsea/index.jsp. Gene sets from papers are referenced in the Supplementary Data 7 Table 3. Source data are provided with this paper. Publicly available scRNA-seq data defining differential genes for myogenesis were obtained from Patel et al. publication Supplementary Table S3[23]. Publicly available ChIP-seq data used in this study are available in the GEO database under accession codes GSE83728 and GSE116344[6,24]. The remaining data are available within the Article, Supplementary Information or Source Data file. Source data are provided with this paper.

## Code availability

Code used is available on Github. https://github.com/CCRGeneticsBranch/scRNAseq_snakemake_pipeline/tree/main and https://github.com/CCRGeneticsBranch/khanlab_pipeline.

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

## Acknowledgements

This project was supported by the Children's Cancer Foundation (YK) and in part with federal funds from the National Cancer Institute, National Institutes of Health, under contract HHSN261200800001E (JK). The content of this publication does not necessarily reflect the views or policies of the Department of Health and Human Services, nor does mention of trade names, commercial products or organizations imply endorsement by the US Government.

## Author contributions

Conceptualization: YK, BEG, JFS, RGH. Methodology: YK, RS, DM, MP, AA, GMW, BZS, JJB, JS, XW, HC, VG, RGH. Investigation: YK, BEG, RS, JFS, AA, SP, GMW, BZS, JJB, JTK, YKS, CW, MO, LMJ, NIT, RGH. Formal analysis: YK, BEG, RS, AA, SP, GMW, DM, JJB, SRS, JRE, HC, VG, MO, LMJ, JSW, BRO, NIT, RGH. Visualization: YK, RS, AA, NIT, HC. Funding acquisition: YK, JK. Resources: AA, GMW, JS, RC, SRS, DE, JJ, LP, MO, JBM, BRO. Project administration: JK. Supervision: RGH, JK. Writing – original draft: YK, RS, DM, JSW, NIT, RGH, JK. Writing – review & editing: YK, DM, BEG, RGH, JK.

## Competing interests

Patent applications for compounds PFI-63 (P3FI-63) and PFI-90 (P3FI-90) have been filed. Applicant: United States of America as represented by the Secretary, Department of Health and Human Services. Inventors: J.K., R.G.H., G.M.W., M.L.P. Application numbers: US 17/777,552; CA 3,158,557; and EP 20824760.1. Status: Pending. Aspects of manuscript covered by patent: The compounds PFI-63 (P3FI-63) and PFI-90 (P3FI-90) are the subject of the patent application. Authors declare that they have no competing interests.

## Additional information

[1]Genetics Branch, NCI, NIH, Bethesda, MD, USA. [2]Department of Genetics and Genome Sciences, Case Western Reserve University, Cleveland, OH, USA. [3]Basic Science Program, Frederick National Laboratory for Cancer Research (FNLCR), Frederick, MD, USA. [4]Pediatric Oncology Branch, NCI, NIH, Bethesda, MD, USA. [5]Collaborative Bioinformatics Resource, NCI, NIH, Bethesda, MD, USA. [6]Department of Hematology and Oncology, Cell and Gene Therapy, Bambino Gesù Children's Hospital, IRCCS, Rome, Italy. [7]Leidos Biomed Res Inc, FNLCR, Basic Sci Program, Frederick, MD, USA. [8]Molecular Targets Program, NCI, NIH, Frederick, MD, USA. [9]Nationwide Children's Hospital, Center for Childhood Cancer Research, Columbus, OH, USA. [10]Department of Pediatrics, The Ohio State University College of Medicine, Columbus, OH, USA. [11]Department of Biological Chemistry & Pharmacology, The Ohio State University College of Medicine, Columbus, OH, USA. [12]Chemical Biology Laboratory, NCI, NIH, Frederick, MD, USA. [13]Genome Modification Core, Laboratory Animal Sciences Program, FNLCR, Frederick, MD, USA. [14]Natural Products Branch, NCI, NIH, Frederick, MD, USA. [15]Protein Expression Laboratory, FNLCR, NIH, Frederick, MD, USA. [16]Protein Characterization Laboratory, FNLCR, NIH, Frederick, MD, USA. [17]Laboratory of Cell Biology, NCI, NIH, Bethesda, MD, USA. [18]Cancer Innovation Laboratory, NCI, NIH, Bethesda, MD, USA. [19]Department of Anatomy and Cell Biology, George Washington University, Washington, DC, USA. ✉ e-mail: yong.kim@nih.gov; khanjav@mail.nih.gov

