## [Peer Review File · Nature Communications]

KDM3B inhibitors disrupt the oncogenic activity of PAX3-FOXO1 in fusion positive rhabdomyosarcoma.Reviewers' Comments:

Reviewer #1:

Remarks to the Author:

Kim et al describe the identification of a compound series that is able to reduce transcriptional activity in a PAX3-FOXO1 reporter cell line, ultimately leading to apoptosis and myogenic differentiation. The PAX3-FOXO1 fusion protein is a key oncogenic driver in fusion-positive alveolar rhabdomyosarcoma, an aggressive childhood malignancy. The authors provide evidence that the main mode of action of the identified compounds might lie in their ability to inhibit the enzymatic activity of histone lysine demethylases (KDMs), an important family of epigenetic modulators. The work describes the screening and target characterisation efforts and includes cellular and in vivo follow-up studies, aiming to support their target hypothesis of an anti-cancer effect of a broad-spectrum KDM inhibitor in rhabdomyosarcoma. As such, this manuscript provides an initial, important step to developing tools for oncology research, by highlighting a possible role of KDMs in fusion-positive rhabdomyosarcoma. There are some compelling arguments in support of their hypothesis, however, at this point, I am hesitant to support the publication of the study in its current format. My criticism is related to the following areas:

* Hit identification from compound screening, biochemical characterisation: the authors performed a compound library screening against a medium-sized collection of chemical matter; the initial hits were further reduced by providing scores and dose-response curves (not exactly described in detail how that was accomplished; cf line 109/110) resulting in the identification of compounds designated PFI63 and PFI90, the main objects of their work. The authors employed Gene Set Enrichment Analysis (GSEA) and identified histone lysine demethylases as possible targets for their identified hits. Whilst this appears reasonable other mechanisms at work might be referred to - for example, other epigenetic manipulations (bromodomain inhibition, PRC2 manipulation such as JQ1, SUZ12) appear as significant hits in their transcriptional analysis, possibly highlighting complex epigenetic changes upon inhibition. At this point, an illustration of differentially regulated genes (heatmap) would be useful to highlight possible mechanisms and transcriptional targets. A direct comparison of their putative KDM inhibitors described in this section (and Discussion) such as JIB-04, GSK-J4, GSK690, and other KDM3 inhibitors) applied to their cell line system with bulk-RNAseq as readout would be useful to corroborate their argument.

Similarity Ensemble Approach (SEA) analysis indeed suggests several 2-oxoglutarate dependent oxygenases (including several KDMs but also FTO) as possible molecular targets apart from several membrane protein mechanisms (which may constitute possible off-targets). At this point in the manuscript, it appears absolutely essential to expand significantly on the possible mode of inhibition using structural analyses. The corresponding Figure 1H allegedly displays the binding of the inhibitor in the active site of KDM3B (the figure needs to be displayed in a larger format). How is this pose determined (and ranked?) – there is no mention of the experimental/computational details except that Pymol is used to create the figure. Minimally the relationships/distances of the inhibitor with key active site residues, cofactor (2-oxoglutarate) as well as active site metal need to be displayed. Although KDM3B appears as one possible target KDM, modelling then should be performed against the other inferred KDMs as well. Of critical consequence is also the alleged inhibition of KDM1A (LSD1), whose 3D structure importantly shows a completely different active site architecture compared to the Fe²⁺/2-oxoglutarate dependent KDMs such as KDM3B. How is inhibition achieved in light of those differences? Moreover, since a lot of the argument lies on proper biochemical characterisation of the KDMs with inhibitors detailed dose-response curves need to be included in their invitro work (only 2 inhibitor concentrations are shown as relative inhibition). What are the actual activities in their biochemical assays? Are these FDH/NADH assays or direct product determination assays using mass spectrometry? These details are not given in their experimental descriptions. Moreover, based on the docking experiments, competition experiments could be designed to determine mode of action (cofactor or substrate inhibition, metal binding etc).

Minor comments related to this section: Do other HDACi than N1302 (used to refine MoA) show overlap with their profiles? Is there a reason why a PhD thesis was cited for the KDM inhibitor GSK-J4

and not the original Kruidenier et al publication? IC50 should be used in connection with inhibitory assays, EC50 in connection with cell-based assays- the authors do not always use it in this way. I would suggest avoiding "drug-like" properties (line 190). Is any metal included in the NMR binding experiments? How do the authors explain higher binding constants (SPR) compared to more potent inhibition, both in cells and against purified protein? It would be good to highlight which constructs have been used for which biochemical assay- it appears to be different (full-length vs truncated KDM).

* Cellular mechanism of action studies: the GSEA analyses support a combined effect of KDM1A and KDM3b (whilst the biochemical analysis for KDM1A appears weak at this point, see above). Is it possible that other components of a KDM1 complex are regulated by KDM3b? Analysis of chromatin states and CHIP-seq data provides useful pieces to the puzzle by highlighting H3K9 methylation as a key factor, in line with the biochemical activity of KDM3b, whereas the contribution of KDM1A to H3K4 levels is less clear. Figure 5 is important to the conclusions, in particular, 5A and B should be enlarged to increase readability and interpretation. Several of the programs used for analysis are not referenced. Are there any KDM1A and KDM3B CHIP-seq data available to look deeper into direct target genes and chromatin locations? It appears that H3K9me changes are significant contributors to the observed phenotypic effects- how is this accomplished? Although H3K9methylation plays a key role in heterochromatin formation, is that really the case in the scenario described? Are there HP1 Chip-seq data available for their system? If heterochromatin formation and silencing of key loci is not the mechanism in their system, what could lead to transcriptional regulation (ie which other chromatin factors with H3K9 binding domains could mediate the observed effects)? Figure 7 with the presumed MoA appears highly hypothetical at this point.

Minor comments related to this section: how was dosage determined in the in vivo experiments? Are there any exposure studies/data for PFI90? In an ideal case studies on primary tumour material would be preferred (in my view) over animal model systems and cell line experiments in order to conclude on clinical/therapeutic utility.

* METHOD section: in general, several of the various Materials and Methods chapters (p 29 ff, submitted manuscript) require serious copy editing in order to improve readability and reproducibility. Much of the biochemical protein work (include structural analysis, docking, KDM assay) is insufficiently or not at all described. Much of the biochemical and medchem KDM foundation was laid by the Structural Genomic Consortium and its academic and private partners (for example construct design, cloning, biochemical and structural characterisation, tool compounds; resources that have been shared with Addgene but also Eurofins) and has not been adequately recognized in this manuscript- for example I would expect mention of the PDB codes used for structural modelling, or mention of the PIs in the cloning and construct design (as stipulated by Addgene on their website how to use/cite the resources). The in vitro assay conditions (largely outsourced to Eurofins) need to be stated. Similarly, the peptide substrate sequence for their in vitro KDM assays needs to be detailed (I could not immediately find it in their submitted manuscript). I could not find in the M&M section how they produce/express affinity tagged constructs for their SPR experiments. There is no mention how the authors performed the docking experiments of their identified compounds.

* Other (minor comments): the authors choose to use PFI* as acronym to label their identified inhibitors. This is unfortunate (and should be changed in my opinion) since the MedChem team at Pfizer (as part of the Structural Genomics Consortium effort to develop chemical tools for epigenetic research) has contributed several well-characterised molecules also labelled PFI-1, PFI-2 etc. It is recommended not to blur the origins of these useful compound resources.

Reviewer #2:

Remarks to the Author:

In this study, Kim and colleagues have screened 62,643 compounds and identified a promising inhibitor PFI-63 for the PAX3-FOXO1 fusion oncogene. By performing RNA-seq, ATAC-seq, docking

analysis and enzymatic assays, they showed that PFI-63 targets histone lysine demethylases (KDMs) with highest selectivity for KDM3B. They conducted structural similarity search to improve the solubility and potency of PFI-63 with a new compound PFI-90. They then analyzed PFI-90 binding to KDM3B using NMR and SPR. Additional *in vitro* and *in vivo* studies further confirmed the effectiveness of PFI-90 in suppressing growth of FP-RMS. Overall, this is a well designed study that identified a potential new drug candidate for treating FP-RMS and a set of *in vitro* and *in vivo* analyses aim to characterize the mechanisms of how PFI-90 works.

The authors may want to consider the following points to improve the manuscript. In particular, as it is important to investigate how inhibition of the general enzymes of KDMs impacts cell growth, the analyses of the histone ChIP-seq and Hi-C data need to be enhanced.

1. KDMs including KDM3B are broadly expressed in many cell types. Supplementary Fig. 2 and 4 showed PFI-63 and PFI-90 inhibition of cell growth in almost all 60 cancer cell lines. Can they also test some normal cell lines and discuss possible side effects? Any symptoms observed in the mice in the *in vivo* experiments?

2. Fig. 5B shows PFI-90 increases H3K4me3 and H3K9me2. Does PFI-90 increase H3K4me3 and H3K9me3 for all genes or specifically for PAX3-FOXO1 target genes? Fig. 5A shows almost no change of ChromHMM states in PAX3-FOXO1 sites. In other words, would inhibition of KDM3B by PFI-90 suppress cell growth by reducing transcription in general? They compared PFI-90 vs. DMSO on PAX3-FOXO1 binding sites. If they compare PAX3-FOXO1 binding sites vs. non-binding sites in PFI-90 treated cells, is there any difference between histone modifications?

3. A technical question on Fig. 5A, why there is almost no signal for H3K9me2 in DMSO even in the heterochromatin state?

4. Fig. 5C shows that the top 10% of enhancers with the most H3K9me2 increase overlaps significantly with PAX3-FOXO1 sites. It'd be helpful to report the total binding sites of all the TFs listed in the figure. Two scenarios should be distinguished: one is that the H3K9me2 specifically increases in PAX3-FOXO1 sites and another is PAX3-FOXO1 has many more binding sites than the other TFs and it is thus natural to have more overlap with the affected enhancers.

5. While H3K27ac intensity does not increase in PAX3-FOXO1 sites, does its intensity increase in all the enhancers? Also, what is the number of H3K27ac peaks in PFI-90 treated cells compared to DMSO?

6. As the histone modifications interplay with each other, KDM3B inhibition may have broad impact on histone modifications. For the ChIP-seq of histone marks, it'd be helpful to identify differentially modified regions between PFI-90 and DMSO. Then check whether the PAX3-FOXO1 sites and target genes are associated with these differentially modified regions.

7. The authors conducted Hi-C experiments to investigate whether chromatin structure is affected by PFI-90 treatment, which is important for understanding how PFI-90 inhibition of KDMs suppresses cell growth. Some additional quantification is needed for the analysis.

First, the similarity of the Hi-C maps between PFI-90 and DMSO should be quantified using such as HiCRep. This will provide a reference to call difference between the two Hi-C experiments.

Second, there are many methods to detect the differences between Hi-C contact maps such as CHES (Galan, *Nat. Genet.*, 2020), Zebra (Vian, *Cell*, 2018), which will give more systematic assessment of the chromatin structure difference.

Third, the chromatin structure difference may become easier to detect while guided by the

differentially modified histone marks. Loop, TAD and compartment comparison can be focused on the regions with differentially modified H3K27ac, H3K9me2, H3K9me3 and H3K4me3.

Fourth, histone modification changes may alter the modularity of the chromatin structure as shown in a recent study (Zheng and Wang, Nat Commons, 2022). It'd be interesting to compare chromatin modularity changes between PFI-90 and DMSO and check how it affects PAX3-FOXO1 target genes and binding sites.

Reviewer #3:

Remarks to the Author:

In their manuscript titled "KDM3B inhibitors disrupt PAX3-FOXO1 oncogenic activity in fusion positive rhabdomyosarcoma," Kim et al. describe a high throughput screen (HTS) using an ARMS cell line engineered to report PAX3-FOXO1 transcriptional activity. As a control, they use a CMV driven reporter line. They identified 64 hits that had specific activity in dose response (DR) and performed in-depth analysis of PFI-63. The molecule altered myogenic differentiation program expression and apoptosis. Timecourse studies were carried out. Another molecule (PFI-90) was identified that had better physical/chemical properties. Within the error for such assays (typically 5-10 fold) these two compounds were equivalent in terms of cellular EC50. The authors went on to show that the molecules inhibited KDM family members with highest potency for KDM3B. However, the IC50 for the enzymes including KDM3B was similar or higher than that for the cells. Binding was analyzed by NMR and SPR and all assays showed similar binding constants (5-10 uM). They went on to perform epigenetic and chromatin profiling and perform dependency screening for KDM3B. They noted that KDM3B was not a dependency in the PedDep portal. They performed limited in vivo testing. Overall, the team set out to identify a PAX3-FOXO1 transcriptional inhibitor and found a small molecule that has modest activity on cells (low micromolar) that they argue inhibits KDM3B. The major concern is the poor binding constant for KDM3B protein and ruling out non-specific effects of the drug on cellular state and survival.

Concerns:

- 1) The EC50 is between 1-10 uM for both compounds and the curve is very steep which raises concerns about specificity of targeting. The authors performed a comprehensive series of studies to try to demonstrate specificity such as NMR and target gene knockdown.
- 2) The enzyme inhibition assays for the KDM family members are in vitro assays and show weak inhibitor activity. This is a major concern. How can the cellular EC50 be similar or lower than the enzyme IC50 activity? I suspect the authors will argue that it inhibits multiple family members and this explains the difference. The concern is that this is a non-specific effect on the cells and that would be consistent with the lack of ARMS selectivity. To address this concern, the authors can:
 - a. Demonstrate direct binding of the protein inside cells.
 - b. Compare their compound to other KDM inhibitors mentioned in the discussion.
 - c. Obtain a crystal structure of the compound bound to protein pocket.
 - d. Make an inactive isomer to distinguish between non-specific chemical properties and selective targeting.
 - e. Test other epigenetic drugs in their reporter assay. How do their inhibitors compare to the clinically relevant epigenetic drugs?
- 3) The drug does not appear to be selective for ARMS. A head to head comparison between ARMS and ERMS using O-PDXs may be beneficial.

Reviewer #4:

Remarks to the Author:

In this manuscript, Kim et al. attempt to identify novel therapeutic targets for fusion-positive alveolar

rhabdomyosarcoma (FP-RMS). Using a small molecule screening approach, they identify one candidate compound as a therapeutic inhibitor of the tumorigenic PAX3-FOXO1 fusion transcript. They then characterize the downstream mechanisms of this compound using multiple complementary approaches, including functional genomics and molecular dynamics simulations, which indicate proteins from the KDM family as putative targets. Finally, they identify a more powerful analog molecule and demonstrate its pre-clinical potential using in vivo xenograft models. The authors generate a wealth of functional genomics data to characterize their candidate compounds, including bulk and single-cell RNA-seq, ChIP-seq, Hi-C, and ATAC-seq data. While the presented results generally align and support the authors' claims, the lack of extensive QC and methodology precludes critical evaluation of some of the results presented here - mainly regarding the functional genomics characterization of the compounds. Therefore, the manuscript will significantly benefit from a comprehensive revision of the methods section and the inclusion of detailed QC for the functional genomics data.

Major comments

* There is a noticeable absence of QC and methodology for all the functional genomics data. The authors should extensively revise their method sections for the functional genomics analyses and provide supplementary figures with QC metrics for each assay. The lack of this information precludes any rigorous evaluation of the methodology used and, consequently, the results presented by the authors. For instance, it is unclear how many libraries were generated for each modality, given that there is no supplementary table detailing the datasets from this study (assay, treatment, time points, sequencing depth, number of cells if applicable, etc.). For example, were the RNA-seq libraries generated in triplicate, quadruplicate, $n=1$? Were technical or biological replicates used? This critical information is missing. Other examples of lacking methods include a description of the EDEN, ChEA, and Enricher analyses (and corresponding references). Similarly, other details are missing. How were *Drosophila* spike-ins added to the sample(s), and how were these reads used to normalize the ChIP-seq data? The same is true for the ATAC-seq data. How were the pre-ranked log₂ fold changes calculated for the ChIP-seq data? How were peak calls performed in the ATAC-seq data? And how were they linked to genes for the GSEA analyses? What genes comprise the gene set induced by treatment with the KDM1A /LSD1 inhibitor SP2509, and where were they obtained? If non-default parameters were used for any software, then the text should describe them or make it explicit that default parameters were used. QC plots should include, for example, MA or volcano plots for RNA-seq data, TSS enrichment plot(s) for ATAC-seq (from *ataqv*, for example).

* I suspect the scRNA-seq clustering captured substantial technical heterogeneity. However, unless the technical sources of variation are extensively characterized and mitigated, it becomes difficult to evaluate the underlying biology described here accurately. For example, solely based on the data presented by the authors, it is not possible to confidently say that the differences in cell number between DMSO and PFI-90-treated cells in cluster 2 are not driven by unaccounted technical variation between the two samples during integration. The threshold of median ± 3 SD for % mitochondrial reads can be very permissive depending on how the data behaves, leading to low-quality barcodes being included. What were the actual cutoffs used for each sample? Assessing this information without additional QC plots and detailed methods is impossible. Based on the methods section, it is unclear if the authors performed clustering before or after removing doublets. Did the authors account for %MT reads and other technical factors as covariates for the SCT normalization? The authors should confirm they are correctly controlling for technical variation by quantifying, for example, the distribution of %MT and nUMI per scRNA-seq cluster.

* Throughout the manuscript, the authors do not clarify several of their analyses in terms of their rationale and interpretation. For instance, the analysis in Figure 1C evaluates the effect of a subset of molecules using gene sets such as "DAVICIONI RMS PAX3/7-FOXO1", etc. In addition to the unquantifiable statements ("few," "manually selected"), no explanation is given in the main text as to what these gene sets correspond, where they were obtained, and how to interpret these results, leaving the reader left to guess the relevance of these gene signatures or search for additional

information outside the main text. Other examples include the "NES" abbreviation in Fig. 3D, which is not explained in the main text or figure legend. In another example, the authors use "leading edge genes" from two signatures (line 342) to compare scRNA-seq clusters. However, there is no explanation of what a "leading edge gene" corresponds to, again leaving the reader guessing its meaning and interpretation.

Minor comments

- * The GSEA results alone are insufficient to compare the transcriptional effects of PFI-90 and KDM knockdowns. Given that these results are directly mentioned in the abstract, the authors should include additional analyses supporting this claim, including plots comparing results at the gene level.
- * Figure text font size in panel 1A should be increased. It seems to be below 5 and requires the reader to zoom in to twice the size to be legible. Please double-check for other small font sizes throughout the figure panels.
- * The Hi-C methods are among the most detailed regarding the functional genomics analyses described in the manuscript, but even that section needs additional details. For example, which annotation was used to define coding genes (line 632)?
- * Figures should have actual p values and FDRs instead of less-than (" $<$ ") shorthands (e.g. Figures 1C, 2E). In addition, all significant results should be clearly labeled to separate them from non-significant results. For example, in Fig. 3D, are all the non-black squares significant?
- * It would be helpful to visualize the ALK super-enhancer luciferase assay for the 26 chemical analogs instead of having these results solely in a table.
- * Line 136: what was the rationale for manually selecting the compounds? Instead of using unquantifiable terms like "[a] few inhibitors," the authors should make statements precisely. How many compounds were selected?
- * NRM and SPR abbreviations should be expanded in the abstract to accommodate the broad Nature Communications readership.
- * Line 575 should read cellranger. Please check for other typos.

REVIEWER COMMENTS

Reviewer #1, expertise in small molecule screening, scRNAseq, epigenetics, histone lysine demethylases (Remarks to the Author):

Kim et al describe the identification of a compound series that is able to reduce transcriptional activity in a PAX3-FOXO1 reporter cell line, ultimately leading to apoptosis and myogenic differentiation. The PAX3-FOXO1 fusion protein is a key oncogenic driver in fusion-positive alveolar rhabdomyosarcoma, an aggressive childhood malignancy. The authors provide evidence that the main mode of action of the identified compounds might lie in their ability to inhibit the enzymatic activity of histone lysine demethylases (KDMs), an important family of epigenetic modulators. The work describes the screening and target characterisation efforts and includes cellular and in vivo follow-up studies, aiming to support their target hypothesis of an anti-cancer effect of a broad-spectrum KDM inhibitor in rhabdomyosarcoma. As such, this manuscript provides an initial, important step to developing tools for oncology research, by highlighting a possible role of KDMs in fusion-positive rhabdomyosarcoma.

We thank the reviewer for the positive comments and the recognition of the importance of KDM function in PAX3-FOXO1 fusion gene driven rhabdomyosarcoma.

There are some compelling arguments in support of their hypothesis, however, at this point, I am hesitant to support the publication of the study in its current format. My criticism is related to the following areas:

* Hit identification from compound screening, biochemical characterisation: the authors performed a compound library screening against a medium-sized collection of chemical matter; the initial hits were further reduced by providing scores and dose-response curves (not exactly described in detail how that was accomplished; cf line 109/110) resulting in the identification of compounds designated PFI63 and PFI90, the main objects of their work.

We apologize for the lack of detail regarding the drug screen. This was a 62,643 compound screen, the largest drug screen to date for rhabdomyosarcoma with consideration of 3 factors for determining PAX3-FOXO1 inhibition. The 3 factors are:

1. PAX3-FOXO1 binding site ALK enhancer driven luciferase as a direct readout for PAX3-FOXO1 transcriptional activity.
2. CMV driven luciferase as a readout for general transcription.
3. XTT assay for cell viability.

The details of the screen have already been published and referenced (Gryder BE, et al 2019. Nat Commun 10, 3004). Given the limited space available, we chose to reference the details instead of describing it. However, we now include additional details in the text (page 4 lines 98-104) that we believe sufficiently describes the screen.

The authors employed Gene Set Enrichment Analysis (GSEA) and identified histone lysine demethylases as possible targets for their identified hits. Whilst this appears reasonable other mechanisms at work might be referred to - for example, other epigenetic manipulations (bromodomain inhibition, PRC2 manipulation such as JQ1, SUZ12) appear as significant hits in their transcriptional analysis, possibly highlighting complex epigenetic changes upon inhibition. At this point, an illustration of differentially regulated genes (heatmap) would be useful to highlight possible mechanisms and transcriptional targets.

We agree with the reviewer that complex epigenetic changes are likely happening. Since we cannot predict how inhibition of a specific enzyme will affect that enzyme's transcript level, it is difficult to directly look at differentially regulated genes to predict possible targets.

Rebuttal Figure 1. Leading Edge Gene analysis from GSEA software comparing PFI-90 vs DMSO at 24 hours. Analysis of gene set to gene set interaction by the fraction of shared leading edge genes. Group 1 are gene sets for PAX3-FOXO1 targets. Groups 2 and 3 have epigenetic gene sets including KDMs, PRC2, BRD4 and HDAC.

Hence, we performed additional analysis using leading edge analysis for P3FI-90 (previously referred to as P3FI-90) using the GSEA desktop software focusing on gene sets with FDR < 0.05. This resulted in a gene set interaction map which showed 3 biologically related groups (Rebuttal Fig 1). The first group was the PAX3-FOXO1 gene sets which, as expected, shared multiple genes. The other groups included epigenetic gene sets for multiple KDMs, PRC2, and BRD gene sets, as you have indicated, which also share multiple genes. Thus, our previous and new analyses highlight these changes occurring across a wide range of important biological pathways that likely include both secondary and tertiary responses to KDM inhibition. Furthermore, many of the differentially expressed genes are regulated by multiple pathways and subsequently across multiple gene sets. Although intriguing, this complex interplay requires extensive experimentation to decipher, and we feel it is outside the scope of this manuscript.

A direct comparison of their putative KDM inhibitors described in this section (and Discussion) such as JIB-04, GSK-J4, GSK-690, and other KDM3 inhibitors) applied to their cell line system with bulk-RNAseq as readout would be useful to corroborate their argument.

Thank you for your suggestion. We performed bulk RNA-seq using JIB-04, GSK-J4, and GSK-690. We found that the multi-KDM inhibitors JIB-04 and GSK-J4 both downregulated PAX3-FOXO1 gene sets robustly. This result is consistent with our findings that KDM inhibition by P3FI-63 (formally known as PFI-63) and P3FI-90 disrupt PAX3-FOXO1 functions. The KDM1A inhibitor GSK-690 was not able to downregulate PAX3-FOXO1 robustly with only 1 downregulated gene set which was related to PAX3-FOXO1. However, GSK-690 was able to upregulate apoptosis gene set. These findings are consistent with our data from CRISPRi knockdown of KDM1A which showed downregulation of PAX3-FOXO1 albeit not as significantly as KDM3B and upregulated apoptosis gene sets. As for testing other KDM3 inhibitors JDI-4 and JDI-12, we were unable to find an easy way to purchase the compounds for testing. Additionally, as discussed in the text, the compounds JDI-4 and JDI-12 compounds were not tested on other KDMs and given the high homology between KDM family members, it is difficult to interpret the data since other KDMs may be inhibited by the JDI compounds. We included these new data in the manuscript as Supplementary Table S1 Tab4-6. Discussion of the results is on page 8 lines 169-174. We thank the reviewer for the suggestions to make our data more robust.

Similarity Ensemble Approach (SEA) analysis indeed suggests several 2-oxoglutarate dependent oxygenases (including several KDMs but also FTO) as possible molecular targets apart from several membrane protein mechanisms (which may constitute possible off-targets). At this point in the manuscript, it appears absolutely essential to expand significantly on the possible mode of inhibition using structural analyses. The corresponding Figure 1H allegedly displays the binding of the inhibitor in the active site of KDM3B (the figure needs to be displayed in a larger format). How is this pose determined (and ranked?) – there is no mention of the experimental/computational details except that Pymol is used to create the figure. Minimally the relationships/distances of the inhibitor with key active site residues, cofactor (2-oxoglutarate) as well as active site metal need to be displayed. Although KDM3B appears as one possible target KDM, modelling then should be performed against the other inferred KDMs as well.

We apologize for the confusion based on original Figure 1H. Figure 1H in the previous version is not a structure based on X-ray crystallography. We did attempt X-ray crystallography to determine the binding of P3FI-63/P3FI-90 to KDM3B. However, these experiments failed to produce crystals, on multiple occasions performed by experts at the NCI. To address the reviewer's concerns, we performed in silico structural docking experiments using ICM-Pro and PoketFinder from Molsoft (details in Methods section). P3FI-90 was tested against KDM3B, KDM4B, KDM5A, KDM5B, KDM6A, KDM6B. We found that P3FI-90 interacts with the metal ions in the active site of KDM3B, KDM4B, KDM5A, KDM5B, KDM6A, but not KDM6B (Fig 2E and Supplementary Figure 7). Hydrogen bond interactions at the active site

showed that KDM3B has the most extensive hydrogen bond interaction and other KDMs had lower interaction which correlates with higher affinity for KDM3B. We also found that P3FI-90 docks well with a favorable score to the active site of LSD1. This is consistent with our inhibition assay which showed inhibition of both KDM3B and LSD1. Lastly, the docking experiment also showed that there may be an alternative binding site for KDM3B for P3FI-90 which indicates that P3FI-90 may have an allosteric inhibitor activity. The model in original Figure 1H has been removed and new structural analysis models have been added to Figure 2E and supplementary figure 7 as well as the Radial Convolutional Neural Net (RTCNN) scores as supplementary table S1, tab8. Discussion of these results are located on pages 9-12 lines 208-255. We thank the reviewer for suggestions to improve our manuscript.

Of critical consequence is also the alleged inhibition of KDM1A (LSD1), whose 3D structure importantly shows a completely different active site architecture compared to the Fe²⁺/2-oxoglutarate dependent KDMs such as KDM3B. How is inhibition achieved in light of those differences?

We agree with the reviewer that this is a critical point. It was a surprise to us when we found that P3FI-90 also inhibited LSD1 given that the 2-oxoglutarate cofactor to Fe²⁺ interaction in the Jumonji family of KDMs are very different from the FAD-dependent action of LSD1. Interestingly, P3FI-90 has significant structural similarity with LSD1 inhibitor SP-2577 (Rebuttal Figure 2).

Rebuttal Figure 2. Structure of P3FI-63, P3FI-90 and SP-2577. SP-2577 structure can be found on <https://pubchem.ncbi.nlm.nih.gov/compound/135565033>.

Additionally, as described above, *in silico* docking experiments show that P3FI-90 is predicted to dock inside the active site of LSD1 favorably (Figure 2E). Finally, we conclusively show by enzyme inhibition assay, clear inhibition of LSD1 by P3FI-90 (Figure 2C). We have included the fact that KDM1A inhibition was a surprise finding on page 9 lines 202-204.

Moreover, since a lot of the argument lies on proper biochemical characterisation of the KDMs with inhibitors detailed dose-response curves need to be included in their *in vitro* work (only 2 inhibitor concentrations are shown as relative inhibition). What are the actual activities in their biochemical assays? Are these FDH/NADH assays or direct product determination assays using mass spectrometry? These details are not given in their experimental descriptions.

We apologize for this omission. As per the reviewer's suggestion, we now have included the details of the biochemical assay performed to determine the

inhibition of enzymes in Figure 1G into the Supplementary Methods section. All of the KDM assays were Time Resolved FRET assays except for KDM1A. KDM1A is an antibody based detection of demethylated product using fluorescence. The Abcam kit (Cat ab113460) is described in the company website and is now included in the manuscript as supplementary methods. As for presenting the entire dose-response, Figure 1G and 2C have been replaced with a new figure incorporating the entire data. The highest concentration tested for each assay was 10 μ M.

Moreover, based on the docking experiments, competition experiments could be designed to determine mode of action (cofactor or substrate inhibition, metal binding etc).

We apologize for the confusion regarding docking experiments as explained above. Original Figure 1H is not X-ray crystallography and therefore does not represent docking of P3FI-90 with KDM3B. We have performed numerous experiments attempting to create crystals with KDM3B and P3FI-63 or P3FI-90 but these have all failed. P3FI-63 was too insoluble and not amenable for crystals. P3FI-90 had better solubility but also did not form crystals. We were able to obtain crystals using NOG as the binding partner as was already published but when attempts were made to soak the crystal with P3FI-90 to replace NOG, the crystals fell apart. We have also attempted to obtain structural information using CryoEM, but only the structured enzymatic pocket was able to be visualized while the disordered portion of KDM3B was too difficult to stabilize for CryoEM. Hence, a competition experiment with resolution to determine docking was not feasible. As shown above, we did perform In Silico Docking experiments which shows that P3FI-90 is predicted to dock at the enzymatic pocket of KDM3B with interaction of the Mn metal. We have included the structural analysis models in Figure 2E and Supplementary Figure 7.

Minor comments related to this section: Do other HDACi than N1302 (used to refine MoA) show overlap with their profiles?

Yes, as you suspected, other HDAC inhibitors such as Spingosine and Bisaprasin also showed the same pattern as N1302 where ALK-Luciferase was decreased while CMV-Luciferase was increased. They are part of the 64 compounds in Table 1 Tab1.

Is there a reason why a PhD thesis was cited for the KDM inhibitor GSK-J4 and not the original Kruidenier et al publication?

We are citing Hookway E to create a list of transcripts downregulated by GSK-J4. Unfortunately, the original Kruidenier et al 2012 Nature paper did not perform a genome wide transcriptome analysis. They performed a PCR based cDNA analysis focusing on the cytokine profile. In Hookway E's thesis, multiple myeloma and Ewing's sarcoma were treated with GSK-J4 and transcriptome analysis was performed using Microarrays. We added this gene set in the GSEA analysis and found it as a hit. We will be happy to remove this if the reviewer requires this.

IC50 should be used in connection with inhibitory assays, EC50 in connection with cell-based assays- the authors do not always use it in this way. I would suggest avoiding "drug-like" properties (line 190).

Thank you for pointing out the inconsistency. We have edited the text for consistency and appropriate usage of EC50 and IC50.

Is any metal included in the NMR binding experiments?

This is an astute question by the reviewer, and we apologize for this omission. For the NMR experiments, Iron was replaced with Zn. We attempted initial experiments with Iron but due to difficulty maintaining the proper ionization

state, we switched to Zn. This approach for Iron based KDMs has already been reported (Leung IK, et al. 2013 Journal of Medicinal Chemistry 56 (2), 547). We have updated the Methods to include this information.

How do the authors explain higher binding constants (SPR) compared to more potent inhibition, both in cells and against purified protein?

SPR Kd and cell free enzyme assay IC50 are in the single digit uM range. Cellular context is more complex and it is not uncommon to have a factor of difference from cell-free assays. Additionally, P3FI-63/P3FI-90 are multi-KDM inhibitors which may have biological effect at lower IC50.

It would be good to highlight which constructs have been used for which biochemical assay- it appears to be different (full-length vs truncated KDM).

Thank you for pointing this out. We have now edited the methods to point out the different KDM3B proteins used for each assay.

* Cellular mechanism of action studies: the GSEA analyses support a combined effect of KDM1A and KDM3b (whilst the biochemical analysis for KDM1A appears weak at this point, see above). Is it possible that other components of a KDM1 complex are regulated by KDM3b?

It is true that many epigenetic factors are part of larger complexes with multiple partners including KDM1A. The hypothesis that KDM3B could be regulating the partner proteins of KDM1A complexes is intriguing. We know from cell free enzyme inhibition assay that KDM1A is directly inhibited by P3FI-90 and the in silico docking experiment showed that P3FI-90 binding to KDM1A had favorable binding score. Since KDM1A is directly inhibited by P3FI-90, teasing out possible indirect inhibition of KDM1A by KDM3B action would be challenging, and we feel is outside the scope of this manuscript. Thus, we feel that showing the strong evidence for inhibition of the primary targets of P3FI-63/P3FI-90 and their biological consequences to FP-RMS biology should be the main scope of this manuscript.

Analysis of chromatin states and CHIP-seq data provides useful pieces to the puzzle by highlighting H3K9 methylation as a key factor, in line with the biochemical activity of KDM3b, whereas the contribution of KDM1A to H3K4 levels is less clear. Figure 5 is important to the conclusions, in particular, 5A and B should be enlarged to increase readability and interpretation.

Thank you for your suggestion. We have now increased the size of both Figure 5A and 5B for easy reading.

Several of the programs used for analysis are not referenced.

We apologize for this omission and have now added references.

Are there any KDM1A and KDM3B CHIP-seq data available to look deeper into direct target genes and chromatin locations? It appears that H3K9me changes are significant contributors to the observed phenotypic effects- how is this accomplished? Although H3K9methylation plays a key role in heterochromatin formation, is that really the case in the scenario described? Are there HP1 Chip-seq data available for their system? If heterochromatin formation and silencing of key loci is not the mechanism in their system, what could lead to transcriptional regulation (ie which other chromatin factors with H3K9 binding domains could mediate the observed effects)?

These are insightful questions. Since these experiments have not been done by us or others in fusion positive rhabdomyosarcoma, we performed ChIP-seq of KDM3B. We

found that the binding of KDM3B correlated well with that of H3K9me2 using deepTools' plotCorrelation (Rebuttal Fig. 3). This was expected, given that H3K9me2 is the target of KDM3B. P3FI-90 treatment did not appreciably change KDM3B binding at TSS or PAX3-FOXO1 sites (See Below, Rebuttal Fig. 4) indicating that P3FI-90 likely inhibits KDM3B's enzymatic function and does not affect its recognition and binding of H3K9me2. We also performed ChIP-seq for HP1 and found that the pattern of binding of HP1 also correlated with H3K9me2 using plotCorrelation (Rebuttal Fig. 3). When looking at co-localization of HP1 and KDM3B, importantly, we found that there was a strong correlation in co-occupancy suggesting that KDM3B binding does indeed take place in heterochromatin. Interestingly, when looking at HP1, P3FI-90 treatment did increase HP1 binding at PAX3-FOXO1 sites. This may indicate that increase in H3K9me2 did increase heterochromatin like characteristics at PAX3-FOXO1 sites by increasing HP1. We have added this to the manuscript (Supplementary Figure 11D-F).

However, our HiC data did not show global changes to the A and B compartments after P3FI-90 treatment. Given that the nearest genes associated with increased H3K9me2 marks were associated with PAX3-FOXO1 ChIP by CHEA analysis in figure 5C, repression of PAX3-FOXO1 target genes by increased H3K9me2 marks may likely act as a dial to tune the PAX3-FOXO1 sites to a more heterochromatin like state without changing compartments. Description of these results can be found on page 24 lines 453-463.

Rebuttal Figure 3. plotCorrelation from the deepTools package using ChIP-seq results from DMSO treated RH4 cells for the following targets: H3K4me3, H3K27ac, H3K9me2, HP1, and KDM3B.

Rebuttal Figure 4. ChIP-seq plotHeatmap profile of KDM3B and HP1 at Transcription Start Site(TSS) and PAX3-FOXO1 sites (P3F) after treatment with P3FI-90 vs DMSO for 24 hours.

We also performed ChIP-seq of KDM1A which showed that its peaks qualitatively co-localized with those in H3K4me3 ChIP-seq experiment (Rebuttal Fig. 5). Due to the high background, peaks calls by MACS2 was limited as can be seen below.

Rebuttal Figure 5. ChIP-seq comparison of H3K4me3 with KDM1A using IGV viewer. Peak calling using MACS2 narrow peak calling set at p-value of 10^{-7} .

Figure 7 with the presumed MoA appears highly hypothetical at this point.

We have modified the discussion to indicate that the theoretical model requires additional studies to confirm the model. However, we believe that this theoretical model helps to create a framework for future discussion and hypothesis testing. This edit is located on page 30 lines 563-564.

Minor comments related to this section: how was dosage determined in the in vivo experiments? Are there any exposure studies/data for PFI90? In an ideal case studies on primary tumour material would be preferred (in my view) over animal model systems and cell line experiments in order to conclude on clinical/therapeutic utility.

We performed drug tolerability study using P3FI-90 at doses 10 mg/kg, 25 mg/kg, and 50 mg/kg. We found that there was significant weight loss and death at the 50 mg/kg dose while 10 mg/kg and 25 mg/kg were well tolerated without weight loss. Hence, for the first in vivo mouse model study, we started treatment at 20 mg/kg

but later, dose was increased to 25 mg/kg which was well tolerated without weight loss.

As for clinically relevant models, we would like to emphasize that P3FI-90 is a tool compound which requires additional Structure Activity Relationships (SAR) studies to make it into a clinical grade drug. We do not claim that P3FI-90 itself will have clinical utility yet and therefore have not added additional PDX models. We have edited the discussion to indicate that additional SAR will be necessary for future clinical utility on page 30 lines 574-577.

* METHOD section: in general, several of the various Materials and Methods chapters (p 29 ff, submitted manuscript) require serious copy editing in order to improve readability and reproducibility. Much of the biochemical protein work (include structural analysis, docking, KDM assay) is insufficiently or not at all described. Much of the biochemical and medchem KDM foundation was laid by the Structural Genomic Consortium and its academic and private partners (for example construct design, cloning, biochemical and structural characterisation, tool compounds; resources that have been shared with Addgene but also Eurofins) and has not been adequately recognized in this manuscript- for example I would expect mention of the PDB codes used for structural modelling, or mention of the PIs in the cloning and construct design (as stipulated by Addgene on their website how to use/cite the resources). The in vitro assay conditions (largely outsourced to Eurofins) need to be stated. Similarly, the peptide substrate sequence for their in vitro KDM assays needs to be detailed (I could not immediately find it in their submitted manuscript). I could not find in the M&M section how they produce/express affinity tagged constructs for their SPR experiments. There is no mention how the authors performed the docking experiments of their identified compounds.

We apologize for these major omissions and thank the reviewer for pointing this out. We have now added more details to the methods section as well as additional references as you have pointed out. Again, we would like to clarify that X-ray crystallography was not part of the manuscript and we apologize for the confusion. We have now completed extensive docking experiments and PDB codes of all structures used have been added. As for the details of the methods from Eurofins experiment, we have added their methods to the supplementary methods section including the peptide substrate information. Lastly, additional details to the KDM3B protein synthesis, expression, and purification have been added to the methods. We thank the reviewers for his/her suggestions with which these modifications have improved our manuscript.

* Other (minor comments): the authors choose to use PFI* as acronym to label their identified inhibitors. This is unfortunate (and should be changed in my opinion) since the MedChem team at Pfizer (as part of the Structural Genomics Consortium effort to develop chemical tools for epigenetic research) has contributed several well-characterised molecules also labelled PFI-1, PFI-2 etc. It is recommended not to blur the origins of these useful compound resources.

Thank you for alerting us to the overlap of our compound names with the Pfizer compounds. We have changed our acronym to P3FI (PAX3-FOXO1 Inhibitor) to avoid this confusion.

Reviewer #2, expertise in Hi-C and 3D genome (Remarks to the Author):

In this study, Kim and colleagues have screened 62,643 compounds and identified a promising inhibitor PFI-63 for the PAX3-FOXO1 fusion oncogene. By performing RNA-seq, ATAC-seq, docking analysis and enzymatic assays, they showed that PFI-63 targets histone lysine demethylases (KDMs) with highest selectivity for KDM3B. They conducted structural similarity search to improve the solubility and potency of PFI-63 with a new compound PFI-90. They then analyzed PFI-90 binding to KDM3B using NMR and SPR. Additional in vitro and in vivo studies further confirmed the effectiveness of PFI-90 in suppressing growth of FP-RMS. Overall, this is a well designed study that identified a potential new drug candidate for treating FP-RMS and a set of in vitro and in vivo analyses aim to characterize the mechanisms of how PFI-90 works.

We thank the reviewer for the positive comments. Please note that PFI-63 and PFI-90 has been changed to P3FI-63 and P3FI-90 as per Reviewer #1 recommendation.

The authors may want to consider the following points to improve the manuscript. In particular, as it is important to investigate how inhibition of the general enzymes of KDMs impacts cell growth, the analyses of the histone ChIP-seq and Hi-C data need to be enhanced.

1. KDMs including KDM3B are broadly expressed in many cell types. Supplementary Fig. 2 and 4 showed PFI-63 and PFI-90 inhibition of cell growth in almost all 60 cancer cell lines. Can they also test some normal cell lines and discuss possible side effects? Any symptoms observed in the mice in the in vivo experiments?

We appreciate the reviewer raised this important question. We performed in vitro testing of primary human fibroblast cell line 7250 and found that EC50 for P3FI-63 was 7.8 μM and P3FI-90 was 1.3 μM (Rebuttal Fig. 6). We believe that KDM3B inhibition leads to increased H3K9me2, which would result in downregulation of transcription. It is likely that 7250 cells have high levels of transcription given their rapid growth rate and hence will be sensitive to P3FI-63 and P3FI-90. However, most importantly, P3FI-90 is well tolerated in the mouse model, at the doses used by us, without significant weight loss at a dose which can delay tumor progression, indicating that the therapeutic index is sufficiently wide enough for a positive response in the tumor. In addition, both P3FI-63 and P3FI-90 are tool compounds as KDM3B-specific inhibitors. We expect additional Structure Activity Relationships (SAR) studies to make them into clinical grade drugs.

Rebuttal Figure 6. EC50 dose response using Incucyte live cell imaging analysis using human primary fibroblast 7250. Drug treatment for 72 hours.

2. Fig. 5B shows PFI-90 increases H3K4me3 and H3K9me2. Does PFI-90 increase H3K4me3 and H3K9me3 for all genes or specifically for PAX3-FOXO1 target genes?

We assume that in the second sentence, the reviewer is referring to H3K9me2 and not H3K9me3 given the context of the first sentence. However, concerning H3K9me3, since KDM3B targets H3K9me2, we predict that H3K9me2 will increase first before H3K9me3 and at the early time point of 24 hours (all ChIP-seq were done at 24 hrs), increase in H3K9me3 may be limited.

As for the increase in H3K4me3 and H3K9me2, we do see a genome wide increase by sequencing read counts (Figure 5B). However, when looking at differential peak calls we see that H3K9me2 is enriched at PAX3-FOXO1 sites and H3K4me3 is enriched at apoptosis genes. To show this, we performed H3K4me3 and H3K9me2 ChIP-seq in triplicates, and calculated the differential peaks using DiffBind. For H3K9me2, we identified 621 upregulated differential peaks and 0 downregulated differential peaks. We determined the nearest genes based on the 621 upregulated peaks and removed duplicate genes resulting in 245 unique genes. This gene list was entered into Enrichr, a web-based analysis tool, for ChIP Enrichment Analysis (ChEA) which is a database of ChIP-seq of transcription factors. ChEA analysis showed that differentially upregulated H3K9me2 peaks were most enriched for PAX3-FOXO1 (Figure 5C). For H3K4me3, we performed the same analysis looking at differential peak analysis using DiffBind. 2391 upregulated peaks were identified and looking at nearest gene and removing duplicate genes, we identified 1,811 unique genes. Since H3K4me3 is an activation mark and RNA-seq showed upregulation of apoptosis, we assessed which biological processes are enriched for H3K4me3 using Gene Ontology for the top 20% upregulated peaks. Many of the top enriched biological processes identified by Gene Ontology were apoptosis pathways consistent with the RNA-seq data. We have clarified this in the manuscript to include these details and can be found on page 20 lines 398-400 and page 23 lines 434-451.

Fig. 5A shows almost no change of ChromHMM states in PAX3-FOXO1 sites. In other words, would inhibition of KDM3B by PFI-90 suppress cell growth by reducing transcription in general? They compared PFI-90 vs. DMSO on PAX3-FOXO1 binding sites. If they compare PAX3-FOXO1 binding sites vs. non-binding sites in PFI-90 treated cells, is there any difference between histone modifications?

The reviewer is likely correct. There is a general increase in read counts for H3K9me2 throughout the genome and it may very well downregulate transcription in general. When we compared H3K9me2 reads at PAX3-FOXO1 sites compared to house-keeping genes (non-PAX3-FOXO1 sites), we found that PFI-90 treatment resulted in 1.9 fold increase over DMSO at PAX3-FOXO1 sites compared to 1.7 fold increase at House Keeping genes (statistically not significant), indicating that H3K9me2 increase is likely general (Rebuttal Fig 7). Since FP-RMS is transcriptionally addicted to PAX3-FOXO1 driven transcription, we believe H3K9me2 increase has a larger impact to PAX3-FOXO1 targets. As for why PAX3-FOXO1 binding is not affected by the increase in H3K9me2, this is likely due to the pioneering effect of PAX3-FOXO1 recently described by the Stanton lab (Sunkel BD, et al 2021 *iScience*, 24(8); 102867).

Rebuttal Figure 7. H3K9me2 ChIP-seq analysis using plotHeatmap at PAX3-FOXO1 sites vs House Keeping genes. Fold increase when comparing P3FI-90 (1uM) treatment vs DMSO for 24 hours.

3. A technical question on Fig. 5A, why there is almost no signal for H3K9me2 in DMSO even in the heterochromatin state?

We apologize for the confusion. The figure is normalized by Gb of state. For example, since 30% of the RH4 chromatin is heterochromatin and active enhancers are only < 1%, we normalized for each of the histone marks by Gb of heterochromatin state and active enhancer state, etc. Below is how the data looks before normalization and you will see that indeed H3K9me2 is enriched in the polychrome and heterochromatin states as expected (Rebuttal Fig 8). However, this figure does not showcase the fact that after P3FI-90 treatment there is disproportionate increase in H3K9me2 at enhancer regions compared to normalized figure.

Rebuttal Figure 8. ChromHMM analysis of histone marks and PAX3-FOXO1. Drosophila spike-in adjusted read counts. No normalization for Gb of states.

4. Fig. 5C shows that the top 10% of enhancers with the most H3K9me2 increase overlaps significantly with PAX3-FOXO1 sites. It'd be helpful to report the total binding sites of all the TFs listed in the figure. Two scenarios should be distinguished: one is that the H3K9me2 specifically increases in PAX3-FOXO1 sites and another is PAX3-FOXO1 has many more binding sites than the other TFs and it is thus natural to have more overlap with the affected enhancers.

As stated above, we have now performed replicate ChIP-seq for H3K9me2 and performed differential peak analysis as triplicates using diffBind (Ross-Innes, C. et al. 2012 Nature 481, 389-393). This produced 621 upregulated differential

peaks and 0 downregulated differential peaks. We determined the nearest genes based on the 621 upregulated peaks and removed duplicate genes resulting in 245 unique genes that showed that PAX3-FOXO1 was the top enriched ChIP-seq profile.

As for the number of genes for each TF in ChEA, we found that the range was between 17 genes to 4931 as below with a mean of 1273 genes (Rebuttal Fig. 9). For PAX3-FOXO1, there are 918 target genes (Cao L, et al Cancer Res. 2010;70(16):6497) making it slightly lower than the average TF. Hence it is unlikely that the enrichment we see for PAX3-FOXO1 is a purely mathematical bias. Biologically however, given that FP-RMS is transcriptionally addicted to PAX3-FOXO1 transcriptional targets, it is likely that any perturbation in transcription will most dramatically affect PAX3-FOXO1 target genes. The number of genes for each ChEA transcription factor has been added to Fig 5C and Supplementary Table S5, Tab2.

Rebuttal Figure 9. ChEA transcription factor ChIP gene sets from the Enricher website <https://maayanlab.cloud/Enrichr/>. Plot of number of genes per each transcription factor.

5. While H3K27ac intensity does not increase in PAX3-FOXO1 sites, does its intensity increase in all the enhancers? Also, what is the number of H3K27ac peaks in PFI-90 treated cells compared to DMSO?

Thank you for this question which is a good way to validate our ChromHMM data. ChromHMM analysis of our ChIP-seq data showed that there is no change in H3K27ac when comparing P3FI-90 treatment vs DMSO. Also, in looking specifically at different enhancer regions, there was also no change in H3K27ac by ChromHMM (Fig. 5A). To validate our ChromHMM results as suggested by this reviewer, we examined if the profile of H3K27ac at all enhancers were different between P3FI-90 vs DMSO. We found that there was no difference in H3K27ac at enhancer sites, consistent with the ChromHMM data (Rebuttal Fig 10).

Rebuttal Figure 10. H3K27ac ChIP-seq analysis using plotHeatmap at all enhancer locations. Comparison between DMSO treated H3K27ac ChIP-seq vs P3FI-90 treated (1uM, 24 hours).

As for the number of H3K27ac peaks, we used MACS2 to call peaks and using a p-value cutoff of 1×10^{-7} , there were 49,398 peaks in P3FI-90 treated cells and 48,180 peaks for DMSO, indicating again no significant change in H3K27ac after P3FI-90 treatment. We have now modified the manuscript to describe these results on page 22 line 423-425 and Supplementary Fig 11A.

6. As the histone modifications interplay with each other, KDM3B inhibition may have broad impact on histone modifications. For the ChIP-seq of histone marks, it'd be helpful to identify differentially modified regions between PFI-90 and DMSO. Then check whether the PAX3-FOXO1 sites and target genes are associated with these differentially modified regions.

To determine differentially modified peaks, we performed replicate ChIP-seq for H3K4me3 and H3K9me2. Then we performed differential peak analysis of the triplicate data using diffBind (Ross-Innes, C. et al. 2012 Nature 481, 389-393). As described in point #4, differential peaks for H3K9me2 were all increased after P3FI-90 treatment with no decreased peaks. ChEA analysis on the differential genes identified PAX3-FOXO1 ChIP as the top hit indicating that increase in H3K9me2 peaks were associated with PAX3-FOXO1 sites (Figure 5C).

7. The authors conducted Hi-C experiments to investigate whether chromatin structure is affected by PFI-90 treatment, which is important for understanding how PFI-90 inhibition of KDMs suppresses cell growth. Some additional quantification is needed for the analysis.

First, the similarity of the Hi-C maps between PFI-90 and DMSO should be quantified using such as HiCRep. This will provide a reference to call difference between the two Hi-C experiments.

We performed HiCRep and did not find significant differences between P3FI-90 and DMSO. We have added this to the manuscript (page 25 line 481-482).

Second, there are many methods to detect the differences between Hi-C contact maps such as CHES (Galan, Nat. Genet., 2020), Zebra (Vian, Cell, 2018), which will give more systematic assessment of the chromatin structure difference.

We also performed CHES and Zebra analysis and they also did not show significant differences. We have added this to the manuscript (page 25 line 481-482).

Third, the chromatin structure difference may become easier to detect while guided by the differentially modified histone marks. Loop, TAD and compartment comparison can be focused on the regions with differentially modified H3K27ac, H3K9me2, H3K9me3 and H3K4me3.

We performed this analysis and found that differential histone marks for H3K9me2 and H3K4me3 did not significantly overlap directly with Loops and TADs. We believe this highlights the fact that overall, the effect on the Loops and TADs were mild as we have already discussed in the manuscript. There were very small changes to the number of loops but the loops did have a lower APA score indicating that the strength of loops decreased.

	Total	H3K4me3 upregulated differential peak sites	H3K9me2 upregulated differential peak sites
diff_TAD_up	1307	171	20
diff_TAD_dn	957	4	7
loops_gained	582	34	5
loops_lost	675	41	4

Table of TAD and Loop analysis delineated by H3K4me3 differential peaks and H3K9me2 differential peaks. Total number of TADs and Loops identified by HiC was used to determine how many of the TADs and Loops overlapped with H3K4me3 and H3K9me2 differential peaks.

Fourth, histone modification changes may alter the modularity of the chromatin structure as shown in a recent study (Zheng and Wang, Nat Commons, 2022). It'd be interesting to compare chromatin modularity changes between PFI-90 and DMSO and check how it affects PAX3-FOXO1 target genes and binding sites.

Modularity of the chromatin from Zheng 2022 Nat Commons paper is determined mostly by H3K27ac histone mark. Our analysis of the cRAM showed that although there was a difference in cRAMs boundaries between PFI90 vs DMSO (Table below), none of the boundaries were enriched for PAX3-FOXO1 target genes. This was expected given that H3K27ac ChIP and ChromHMM showed no difference between DMSO vs P3FI-90.

	DMSO only	PFI90 only	Overlap	Total
H3K27ac cRAM boundaries	74	71	821	966
PAX3-FOXO1 Target Gene Set FDR	FDR = 1	FDR = 1	FDR = 0.59	

Table of cRAM boundaries that are found in DMSO treated only, P3FI-90 treated only, and in both conditions. Corresponding PAX3-FOXO1 target gene set enrichment at cRAM boundaries.

Reviewer #3, expertise in rhabdomyosarcoma epigenomics and models, RNAseq, ChIP-seq, high throughput screening and drug discovery (Remarks to the Author):

In their manuscript titled "KDM3B inhibitors disrupt PAX3-FOXO1 oncogenic activity in fusion positive rhabdomyosarcoma," Kim et al. describe a high throughput screen (HTS) using an ARMS cell line engineered to report PAX3-FOXO1 transcriptional activity. As a control, they use a CMV driven reporter line. They identified 64 hits that had specific activity in dose response (DR) and performed in-depth analysis of PFI-63. The molecule altered myogenic differentiation program expression and apoptosis. Timecourse studies were carried out. Another molecule (PFI-90) was identified that had better physical/chemical properties. Within the error for such assays (typically 5-10 fold) these two compounds were equivalent in terms of cellular EC50. The authors went on to show that the molecules inhibited KDM family members with highest potency for KDM3B. However, the IC50 for the enzymes including KDM3B was similar or higher than that for the cells. Binding was analyzed by NMR and SPR and all assays showed similar binding constants (5-10 uM). They went on to perform epigenetic and chromatin profiling and perform dependency screening for KDM3B. They noted that KDM3B was not a dependency in the PedDep portal. They performed limited in vivo testing. Overall, the team set out to identify a PAX3-FOXO1 transcriptional inhibitor and found a small molecule that has modest activity on cells (low micromolar) that they argue inhibits KDM3B. The major concern is the poor binding constant for KDM3B protein and ruling out non-specific effects of the drug on cellular state and survival.

Please note that PFI-63 and PFI-90 has been changed to P3FI-63 and P3FI-90 as per Reviewer #1 recommendation.

Concerns:

1) The EC50 is between 1-10 uM for both compounds and the curve is very steep which raises concerns about specificity of targeting. The authors performed a comprehensive series of studies to try to demonstrate specificity such as NMR and target gene knockdown.

We agree that the curves are steep for some of our cell lines, however as shown in the testing of the NCI60 cell lines, this is not the case for all cell lines, arguing against non-specific toxicity. We also show specificity using enzymatic assays and NMR analysis. We acknowledge that this is a tool compound requiring additional Structure Activity Relationships (SAR) studies to make it into a clinical grade drug.

2) The enzyme inhibition assays for the KDM family members are in vitro assays and show weak inhibitor activity. This is a major concern. How can the cellular EC50 be similar or lower than the enzyme IC50 activity? I suspect the authors will argue that it inhibits multiple family members and this explains the difference. The concern is that this is a non-specific effect on the cells and that would be consistent with the lack of ARMS selectivity. To address this concern, the authors can:

a. Demonstrate direct binding of the protein inside cells.

Although these experiments would add to the evidence of binding between P3FI-90 and KDM3B, the experiments are technically difficult to execute. However, the data of In Silico Docking, SPR, and NMR are compelling to show specific binding of P3FI-90 to KDM3B. In addition, the biological result of increased methylation of histones further validate that P3FI-90 does target KDM3B and KDM1A specifically.

b. Compare their compound to other KDM inhibitors mentioned in the discussion.

Thank you for your suggestion. We performed bulk RNA-seq using JIB-04, GSK-J4, and GSK-690. We found that the multi-KDM inhibitors JIB-04 and GSK-J4 both downregulated PAX3-FOXO1 gene sets robustly. This result is consistent with our findings that KDM inhibition by P3FI-63 and P3FI-90 can downregulate PAX3-FOXO1. The KDM1A inhibitor GSK-690 was not able to downregulate PAX3-FOXO1 robustly with only 1 downregulated gene set related to PAX3-FOXO1. However, GSK-690 was able to upregulate apoptosis gene set. These findings are consistent with our data from CRISPRi knockdown of KDM1A which also showed downregulation of PAX3-FOXO1 albeit not as significantly as KDM3B and upregulated apoptosis gene sets. The new results have been added to Supplementary Table 1 Tab 4-6 and results presented on page 8 lines 169-174.

c. Obtain a crystal structure of the compound bound to protein pocket.

We have performed numerous experiments attempting to create crystals with KDM3B and P3FI-63 or P3FI-90 without success. P3FI-63 was too insoluble and not amenable for crystals. P3FI-90 had better solubility but also did not form crystals. We were able to obtain crystals using NOG as the binding partner as published. But when attempts were made to soak the crystal with P3FI-90 to replace NOG, the crystal fell apart. We have also attempted to obtain structural information using CryoEM. Only the structured enzymatic pocket was able to be visualized, and the disordered portion of KDM3B was too difficult to stabilize for CryoEM. We have now performed In Silico Docking experiments which predict binding of P3FI-90 to KDM3B, KDM1A, KDM4B, KDM5A, KDM5B, KDM6A, KDM6B. We found that KDM3B has favorable binding score by Radial Convolutional Neural Net (RTCNN) score. Also, P3FI-90 had the most extensive hydrogen bond interaction with KDM3B. We also found that P3FI-90 had favorable binding score for KDM1A. We have included these results in Fig. 2E, Supplementary Figure 7, Supplementary Table S1 Tab8 and the results presented on page 9 lines 209-255. Thank you for this suggestion and we believe the addition of the in silico docking experiment strengthens our study.

d. Make an inactive isomer to distinguish between non-specific chemical properties and selective targeting.

We have shown that P3FI-90 is a multi KDM inhibitor with highest activity against KDM3B, and have demonstrated specificity by enzymatic assays, NMR, and changes in histone modification. We believe manufacturing an inactive isomer is outside the scope of this study.

e. Test other epigenetic drugs in their reporter assay. How do their inhibitors compare to the clinically relevant epigenetic drugs?

In our previous paper (Gryder BE, et al. 2019 Nat Commun 10, 3004), we explored this question using groups of well characterized epigenetic drugs (Rebuttal Fig 11 below). In that publication, we focused on bromodomain inhibitors and HDAC inhibitors given a strong signal in downregulating PAX3-FOXO1 driven ALK-

luciferase by multiple drugs in the same class. The signal from KDMs was weaker and only a few KDMs were tested in this study. In our current submission, we started with a novel compound of unknown mechanism of action and identified its targets to be KDM3B and KDM1A. Of note, even for HDACi and BRDi, majority of compounds were only effective at single digit μM or higher which is comparable to P3FI-63 and P3FI-90, indicating KDM inhibition as an effective target for FP-RMS. We appreciated insight of this reviewer, and we believe the suggestions have helped us to improve the manuscript.

Rebuttal Figure 11. Result of drug screen using RH4 cell lines modified with ALK super enhancer PAX3-FOXO1 binding driven Luciferase, CMV driven Luciferase, and XTT assay. Drugs categorized by type of response: PAX3-FOXO1 Super Enhancer selective, ALK-Luc down CMV-Luc up (or HDAC-like) response, or non-specific.

3) The drug does not appear to be selective for ARMS. A head to head comparison between ARMS and ERMS using O-PDXs may be beneficial.

Thank you for your assessment. We have provided data in Supplemental Figure 1D that sensitivity for ERMS was in the single digit μM range similar to that of ARMS. We have included in the discussion that increase in H3K9me2 by inhibition of KDM3B is not specifically targeting PAX3-FOXO1 targets but that more likely, it is a general transcriptional downregulation which disproportionately affects PAX3-FOXO1 targets. We agree with the reviewers that this drug is not a selective inhibitor for ARMS and may have cytostatic/cytotoxic activities in other cancer types since arguably all cancers have lineage-specific transcriptional dependencies for survival. Despite showing activity in multiple cell types, the predominant effect on ARMS cells is inhibiting the essential PAX3-FOXO1 co-regulatory networks that are unique to this cancer. Further studies will be needed to determine if master transcription factor networks are similarly disrupted in other cancer types.

Reviewer #4, expertise in ATAC-seq (Remarks to the Author):

In this manuscript, Kim et al. attempt to identify novel therapeutic targets for fusion-positive alveolar rhabdomyosarcoma (FP-RMS). Using a small molecule screening approach, they identify one candidate compound as a therapeutic inhibitor of the tumorigenic PAX3-FOXO1 fusion transcript. They then characterize the downstream mechanisms of this compound using multiple complementary approaches, including functional genomics and molecular dynamics simulations, which indicate proteins from the KDM family as putative targets. Finally, they identify a more powerful analog molecule and demonstrate its pre-clinical potential using in vivo xenograft models. The authors generate a wealth of functional genomics data to characterize their candidate compounds, including bulk and single-cell RNA-seq, ChIP-seq, Hi-C, and ATAC-seq data. While the presented results generally align and support the authors' claims, the lack of extensive QC and methodology precludes critical evaluation of some of the results presented here - mainly regarding the functional genomics characterization of the compounds. Therefore, the manuscript will significantly benefit from a comprehensive revision of the methods section and the inclusion of detailed QC for the functional genomics data.

Please note that PFI-63 and PFI-90 has been changed to P3FI-63 and P3FI-90 as per Reviewer #1 recommendation.

Major comments

* There is a noticeable absence of QC and methodology for all the functional genomics data. The authors should extensively revise their method sections for the functional genomics analyses and provide supplementary figures with QC metrics for each assay. The lack of this information precludes any rigorous evaluation of the methodology used and, consequently, the results presented by the authors. For instance, it is unclear how many libraries were generated for each modality, given that there is **no supplementary table detailing the datasets from this study (assay, treatment, time points, sequencing depth, number of cells if applicable, etc.)**. For example, were the RNA-seq libraries generated in triplicate, quadruplicate, n=1? Were technical or biological replicates used? This critical information is missing. Other examples of lacking methods include a description of the EDEN, ChEA, and Enricher analyses (and corresponding references). Similarly, other details are missing. How were Drosophila spike-ins added to the sample(s), and how were these reads used to normalize the ChIP-seq data? The same is true for the ATAC-seq data.

We apologize for the insufficient inclusion of QCs and other experimental details. We have included additional details in the methods and in the Supplementary Table 7 Tab2. References have also been added.

For the Drosophila spike-in, we followed the manufacturer's recommendations for the ChIP-IT High Sensitivity Kit (Active Motif). 50 ng of spike-in chromatin is added to each ChIP sample and 2 ug of ChIP antibody is added for pull down along with the RH4 chromatin and the anti-human antibody. Once the sequencing is completed, we normalized the ChIP-seq data by using a scaling factor $10^6 / (\text{the read count of Drosophila reads})$. These details have been added to the methods.

As for the ATAC-seq, there was no spike-in. We apologize for the mistake in the text. It has now been edited.

How were the pre-ranked log₂ fold changes calculated for the ChIP-seq data? How were peak calls performed in the ATAC-seq data? And how were they linked to genes for the GSEA analyses?

The Pre-ranked log₂ fold changes were calculated as follows and was the same process for ATAC. We merged the DMSO and PFI90 ChIP(or ATAC) peaks using "bedtools merge". Then we used multiBigwigSummary from deepTools to calculate the count coverage of these merged peaks. To identify the nearest genes, we used annotatePeaks.pl from HOMER suite to annotate the peaks. Finally, we calculated the log₂ fold changes for the nearest peak for each gene to generate the GSEA rank file. These details have been added to the methods and additional details are also located in Supplementary Table 7, tab2.

For ATAC-seq peak calling, we used MACS2 narrow peak calling using p-value = 10^{-7} . In fact, all narrow peaks used the same p-value.

What genes comprise the gene set induced by treatment with the KDM1A /LSD1 inhibitor SP2509, and where were they obtained?

The gene set for SP2509 represents 103 genes induced in Ewing's sarcoma cells following treatment with KDM1A/LSD1 inhibitor SP2509 (from Fig. 5A of Pishas et al., 2018 Mol Cancer Ther; 17 (9): 1902–1916). The GMT file used for GSEA has been annotated and references indicated in the description. The GMT file information has been added to Supplementary Table S7 tab3.

If non-default parameters were used for any software, then the text should describe them or make it explicit that default parameters were used. QC plots should include, for example, MA or volcano plots for RNA-seq data, TSS enrichment plot(s) for ATAC-seq (from ataqv, for example).

We have added additional language to our Methods section to explicitly state which parameters were used in our data analyses. Furthermore, for analyses where non-default parameters were used and are important for the interpretation of our results, we explicitly mention these changes.

These QC plots can be found in Supplementary Figure 14 A-C and are referenced in the methods section.

* I suspect the scRNA-seq clustering captured substantial technical heterogeneity. However, unless the technical sources of variation are extensively characterized and mitigated, it becomes difficult to evaluate the underlying biology described here accurately. For example, solely based on the data presented by the authors, it is not possible to confidently say that the differences in cell number between DMSO and PFI-90-treated cells in cluster 2 are not driven by unaccounted technical variation between the two samples during integration. The threshold of median ± 3 SD for % mitochondrial reads can be very permissive depending on how the data behaves, leading to low-quality barcodes being included. What were the actual cutoffs used for each sample? Assessing this information without additional QC plots and detailed methods is impossible. Based on the methods section, it is unclear if the authors performed clustering before or after removing doublets.

Thank you for carefully assessing our data for accuracy and quality. We used the standard cutoff of > 200 features/cell. We also determined the mitochondrial % in each cluster and found that mitochondrial % were $< 15\%$ in all clusters. Cluster 3 did have lower features compared to other clusters but still maintained the standard cutoff of > 200 features/cell. Additionally, MT% was proportionately lower in Cluster 3 which was reassuring. Also, biologically, it is consistent with the fact that cluster 3 has low PAX3-FOXO1 and low Myoblast signatures which would decrease proliferation and transcription resulting in the low features/cell. QC plots with features/cell and mitochondria percent has been added as Supplementary Figure 10 B-C. For the analysis, doublets were identified and removed before clustering. The analysis pipeline is now published on Github and reference in the paper which we hope will clarify our methodology.

https://github.com/CCRGeneticsBranch/scRNAseq_snakemake_pipeline/tree/main

Did the authors account for %MT reads and other technical factors as covariates for the SCT normalization?

Individual sample normalization was done using SCTransform through Seurat V3 setting the variable.features.rv.th to 1.3. No other factors including MT reads were used for normalizations. This detail has been added to the methods. Additionally, as above, we have published our pipeline on GitHub which can be viewed for clarification.

The authors should confirm they are correctly controlling for technical variation by quantifying, for example, the distribution of %MT and nUMI (nCount) per scRNA-seq cluster.

We have now included a QC figure which contains the %MT and nCount per cluster to the manuscript (Supplementary Figure 10B and 10C).

* Throughout the manuscript, the authors do not clarify several of their analyses in terms of their rationale and interpretation. For instance, the analysis in Figure 1C evaluates the effect of a subset of molecules using gene sets such as "DAVICIONI RMS PAX3/7-FOXO1", etc. In addition to the unquantifiable statements ("few," "manually selected"), no explanation is given in the main text as to what these gene sets correspond, where they were obtained, and how to interpret these results, leaving the reader left to guess the relevance of these gene signatures or search for additional information outside the main text. Other examples include the "NES" abbreviation in Fig. 3D, which is not explained in the main text or figure legend. In another example, the authors use "leading edge genes" from two signatures (line 342) to compare scRNA-seq clusters. However, there is no explanation of what a "leading edge gene" corresponds to, again leaving the reader guessing its meaning and interpretation.

We have made clarifying changes in the text based on these suggestions and we think it will improve the interpretability for readers. We thank the reviewer for their comment.

For the GSEA analysis gene sets, the GMT file containing the gene set has been included as a Supplemental Table 7 tab 3, with annotation to indicate the reference paper or GSEA website link from where the gene set came from.

As for the novel compound selection for RNA-seq analysis, we have removed the words "few" and "manually selected". They have been replaced with a more descriptive statement: "To investigate potential mechanisms of action of novel compounds, we selected 4 HDAC-like, 2 ALK-selective, and 1 strong inhibitor with diverse molecular structures for RNA-seq analysis."

NES refers to Normalized Enrichment Score, and this and other acronyms such as FDR has been now defined in the Figure 1 legend or where they first appear.

The leading-edge subset in a gene set are those genes that appear in the ranked list at or before the point at which the running sum reaches its maximum deviation from zero (PMID: 16199517). The leading-edge subset can be interpreted as the core that accounts for the gene set's enrichment signal and provides insights into the underlying biology and pinpoint the most relevant genes in a gene set for a given phenotype. Leading-edge genes used were from Cluster 2 GSEA analysis looking at Myoblast (Supplementary Table S4, tab1), Myocyte (Supplementary Table S4, tab1), and PAX3-FOXO1 (GRYDER_PAX3-FOXO1_CRC Gene Set, Supplementary Table S7, tab3). These genes were used to plot the heatmap (Supplementary Fig 10A). The details have been added to the methods section.

Minor comments

* The GSEA results alone are insufficient to compare the transcriptional effects of PFI-90 and KDM knockdowns. Given that these results are directly mentioned in the abstract, the authors should include additional analyses supporting this claim, including plots comparing results at the gene level.

Thank you for the suggestion on further analysis of the RNA-seq data. To assess the KDM knockdown at the gene level, we used the entire protein coding transcriptome and performed hierarchical clustering analysis with heatmap using heatmap.2 from the gplots package (Rebuttal Fig 12). As can be seen below, P3FI-90 clustered closer to the JumjC family of KDMs (KDM3B, KDM4B, and KDM5A) than to

KDM1A which is not part of the JumjC family. It is likely that the clustering is picking up the similarities in the JumjC family KDM vs non-JumjC. Additionally, P3FI-90 had the highest inhibition of KDM3B and P3FI-90's clustering with the JumjC family is likely driven by the highest inhibition of KDM3B. Gene level analysis using hierarchical clustering was not able to determine which combination of KDM knockdowns was most closely matched by P3FI-90.

Rebuttal Figure 12. Heatmap with Hierarchical clustering using all protein coding genes. Log 2 fold change vs control CRISPRi was used to generate the figure.

Hence, we decided to analyze the gene level data in the reverse starting with P3FI-90 genes. We determined the Log 2 Fold Changed and used the top 50 upregulated genes to create a Gene Set called P3FI90_UP (now added to Supplementary table S7, tab 3). We also took the top 50 downregulated genes to create a Gene Set called P3FI90_DOWN (now added to Supplementary table S7, tab 3). Once the gene sets were created, we repeated the GSEA analysis to determine the enrichment scores. We reasoned that the KMD knockdown closest to P3FI-90 would be most enriched for the gene sets. We found that for P3FI90_DOWN, KDM3B knockdown had the most negative NES followed by KDM1A+KDM3B (Rebuttal Fig 13). When looking at P3FI90_UP, KDM1A+KDM3B combination had the most positive NES. Overall, we can see that KDM1A+KDM3B combination best reflects P3FI-90 gene sets. This new analysis has been added as Fig 3E-F and the results are described on page 17 line 340-348.

Rebuttal Fig 13. New gene sets were created using the top upregulated and top downregulated genes from P3FI-90 called P3FI90_UP and P3FI90_DOWN. GSEA analysis result showing NES by each KDM knockdown.

* Figure text font size in panel 1A should be increased. It seems to be below 5 and requires the reader to zoom in to twice the size to be legible. Please double-check for other small font sizes throughout the figure panels.

We have increased the font size to readability.

* The Hi-C methods are among the most detailed regarding the functional genomics analyses described in the manuscript, but even that section needs additional details. For example, which annotation was used to define coding genes (line 632)?

We have included additional details to the methods section including that we used the GRCh37(hg19) RefSeq annotation to define coding genes.

* Figures should have actual p values and FDRs instead of less-than (" $<$ ") shorthands (e.g. Figures 1C, 2E). In addition, all significant results should be clearly labeled to separate them from non-significant results. For example, in Fig. 3D, are all the non-black squares significant?

We have edited the figures to have actual values instead of " $<$ " shorthand. We have also labeled all significant findings including Figure 3D.

* It would be helpful to visualize the ALK super-enhancer luciferase assay for the 26 chemical analogs instead of having these results solely in a table.

We have added a supplementary figure 4 with the 26 analogs to help view the assay more easily.

* Line 136: what was the rationale for manually selecting the compounds? Instead of using unquantifiable terms like "[a] few inhibitors," the authors should make statements precisely. How many compounds were selected?

Our strategy was to use a series of filtering criteria (i.e. specific ALK-luciferase inhibition) to compile a short list of candidate compounds which had the most favorable characteristics from our screen. From this final list of 39 novel compounds, we elected to evaluate compounds which had a diversity of structural properties rather than continuing selecting based on metrics like score rank. We selected 7 compounds in the first group with 4 from HDAC-like, 2 from ALK-selective, and 1 from Strong Inhibitor and evaluated them using RNA-seq to identify if any of these compounds showed strong inhibition of PAX3-FOXO1 gene signatures, a broader representation of PAX3-FOXO1 function than our ALK-reporter system. The selected compound P3FI-63 was the first compound where the RNA-seq profile showed a strong downregulation of PAX3-FOXO1 gene sets and hence this compound was characterized. The other 39 compounds will be investigated by RNA-seq in future studies. The manuscript has been edited to reflect these changes (page 4 lines 114-129).

* NRM and SPR abbreviations should be expanded in the abstract to accommodate the broad Nature Communications readership.

We have expanded the NMR and SPR abbreviations in the manuscript.

* Line 575 should read cellranger. Please check for other typos.

We have corrected this error and checked for other typos.

Reviewers' Comments:

Reviewer #1:

Remarks to the Author:

My previous comments have been adequately addressed. I think the authors did a great job of addressing the concerns and the manuscript has clearly improved. I still think the reference to Dr Ed Hookway's thesis can be omitted- not because it is wrong but because it might pose difficulties for other researchers in obtaining the content.

Reviewer #2:

Remarks to the Author:

The authors have made plausible efforts to conduct new analyses and experiments to address this reviewer's concerns. The authors may want to include the EC50 of P3FI-63 and P3FI-90 in 7250 cell line and the discussion of the mouse phenotype in the rebuttal letter (better specify the dose and mouse weight change) in the manuscript.

Reviewer #3:

Remarks to the Author:

The reviewers have done a good job of addressing my concerns.

Reviewer #4:

Remarks to the Author:

The authors addressed the majority of my requests appropriately. I appreciate the effort to improve the methods descriptions. From a reproducibility point of view, the manuscript is much more robust. Regarding the scRNA-seq analyses, including the QC metrics helps evaluate the authors' findings. There are noticeable differences in features traditionally used for QC (number of expressed genes and read depth) in some clusters (particularly cluster 2), which would make one wary of technical artifacts. However, because P3FI-90 likely interferes with cell biology at the fundamental level of nuclear architecture, the interpretation of these metrics becomes much more nuanced. The flow cytometry results support the scRNA-seq results, making me agree with the authors' interpretation. Overall, I only have minor requests and suggestions to facilitate readability.

Minor comments

* Table S7 tab 2 addresses my concerns regarding the description and interpretation of the gene sets. To facilitate this information reaching the reader, I suggest the authors reference this table when discussing the GSEA analyses in the main text. If this breaks the order of the tables, instead include a statement along the lines of "Detailed information about the gene sets used for GSEA analyses can be found in the Methods section."

* The figures have plenty of whitespace, and most font sizes are unnecessarily small. I suggest the authors use this space to make font sizes larger in axis labels when possible. For example, the revised Fig. 1A font size is still tiny. Fig. 2C's X- and Y-axis labels could be the same size as 1B. In fact, if the authors can make all axis labels the same size as 1B, the manuscript's readability would improve dramatically. As for the color labels, 1B's are also very small - consider increasing color label sizes to enhance visibility, similar to the size in Figure 1G, for a more consistent presentation across all figures.

* As an unsolicited piece of advice for future reviews, the authors should consider including a version of their manuscript with changes highlighted to facilitate the review process. This practice enhances

clarity for reviewers and streamlines the evaluation process. In the particular case of this review, the authors also had several rebuttal figures and additional explanations. Still, it wasn't always clear to me what was rebuttal-only content and what was updated in the manuscript. I had to use a document comparison tool to determine what was changed between versions and what materials were specific to the rebuttal document.

Reviewer #1 (Remarks to the Author):

My previous comments have been adequately addressed. I think the authors did a great job of addressing the concerns and the manuscript has clearly improved. I still think the reference to Dr Ed Hookway's thesis can be omitted- not because it is wrong but because it might pose difficulties for other researchers in obtaining the content.

Thank you for your comments. We feel that by addressing your concerns, the paper was substantially strengthened, and we appreciate your feedback. We understand your concern and to make the thesis easy to find, we have provided the URL from the Oxford University Research Archive site to the Reference as well as in the GSEA supplemental table 7 tab 2.

<https://ora.ox.ac.uk/objects/uuid:b591861f-985b-4722-8027-492e750f3ff7>

Reviewer #2 (Remarks to the Author):

The authors have made plausible efforts to conduct new analyses and experiments to address this reviewer's concerns. The authors may want to include the EC50 of P3FI-63 and P3FI-90 in 7250 cell line and the discussion of the mouse phenotype in the rebuttal letter (better specify the dose and mouse weight change) in the manuscript.

Thank you for your comments. We feel that by addressing your concerns, the paper was substantially strengthened, and we appreciate your feedback. We have included the EC50 results of 7250 cell line in the supplementary figure 1D and in the text of the Results section. Also, we have added a description of the mouse phenotype to the Discussion section.

Reviewer #3 (Remarks to the Author):

The reviewers have done a good job of addressing my concerns.

Thank you for accepting our manuscript. We feel that by addressing your concerns, the paper was substantially strengthened, and we appreciate your feedback.

Reviewer #4 (Remarks to the Author):

The authors addressed the majority of my requests appropriately. I appreciate the effort to improve the methods descriptions. From a reproducibility point of view, the manuscript is much more robust. Regarding the scRNA-seq analyses, including the QC metrics helps evaluate the authors' findings. There are noticeable differences in features traditionally used for QC (number of expressed genes and read depth) in some clusters (particularly cluster 2), which would make one wary of technical artifacts. However, because P3FI-90 likely interferes with cell biology at the fundamental level of nuclear architecture, the interpretation of these metrics becomes much more nuanced. The flow cytometry results support the scRNA-seq results, making me agree with the authors' interpretation. Overall, I only have minor requests and suggestions to facilitate readability.

Minor comments

* Table S7 tab 2 addresses my concerns regarding the description and interpretation of the gene sets. To facilitate this information reaching the reader, I suggest the authors reference this table when discussing the GSEA analyses in the

main text. If this breaks the order of the tables, instead include a statement along the lines of "Detailed information about the gene sets used for GSEA analyses can be found in the Methods section."

Thank you for your comments. We have included the statement "Detailed information about the gene sets used for GSEA analyses can be found in the Methods section."

* The figures have plenty of whitespace, and most font sizes are unnecessarily small. I suggest the authors use this space to make font sizes larger in axis labels when possible. For example, the revised Fig. 1A font size is still tiny. Fig. 2C's X- and Y-axis labels could be the same size as 1B. In fact, if the authors can make all axis labels the same size as 1B, the manuscript's readability would improve dramatically. As for the color labels, 1B's are also very small - consider increasing color label sizes to enhance visibility, similar to the size in Figure 1G, for a more consistent presentation across all figures.

Thank you for pointing out the readability issues in the figures. We have increased the size of the fonts to aid in the readability of the figures.

* As an unsolicited piece of advice for future reviews, the authors should consider including a version of their manuscript with changes highlighted to facilitate the review process. This practice enhances clarity for reviewers and streamlines the evaluation process. In the particular case of this review, the authors also had several rebuttal figures and additional explanations. Still, it wasn't always clear to me what was rebuttal-only content and what was updated in the manuscript. I had to use a document comparison tool to determine what was changed between versions and what materials were specific to the rebuttal document.

Thank you for your suggestion. We tried to facilitate locating the edited sections by providing line numbers but in retrospect, having the changes highlighted would have been easier for reviewers. Additionally pointing out which content are for rebuttal only would have been helpful as you point out. We will keep these suggestions in mind in the future.